# Crustal permeability generated through microearthquakes is constrained by seismic moment

Pengliang Yu [1,2] ✉, Ankur Mali [3], Thejasvi Velaga[4], Alex Bi[5], Jiayi Yu[2], Chris Marone[1,6], Parisa Shokouhi[7] & Derek Elsworth [1,2] ✉

We link changes in crustal permeability to informative features of microearthquakes (MEQs) using two field hydraulic stimulation experiments where both MEQs and permeability evolution are recorded simultaneously. The Bidirectional Long Short-Term Memory (Bi-LSTM) model effectively predicts permeability evolution and ultimate permeability increase. Our findings confirm the form of key features linking the MEQs to permeability, offering mechanistically consistent interpretations of this association. Transfer learning correctly predicts permeability evolution of one experiment from a model trained on an alternate dataset and locale, which further reinforces the innate interdependency of permeability-to-seismicity. Models representing permeability evolution on reactivated fractures in both shear and tension suggest scaling relationships in which changes in permeability ($\Delta k$) are linearly related to the seismic moment ($M$) of individual MEQs as $\Delta k \propto M$. This scaling relation rationalizes our observation of the permeability-to-seismicity linkage, contributes to its predictive robustness and accentuates its potential in characterizing crustal permeability evolution using MEQs.

The distribution of permeabilities in the shallow crust are known to diminish as a power law with depth[1–3]. This is driven by both the extreme sensitivity of fracture permeability to increasing stress[4] and the rapidity with which damage occasioned by tectonic strains will heal and seal[5–8]. Both stress and temperatures increase with depth. The attempt to create a fluid transmissive crust for the recovery of energy or fuels typically relies on reactivating existing fractures in shear[9] or fracturing in tension[10,11] – each mode of hydraulic-shearing or hydraulic-fracturing driven by artificially elevated fluid pressures. These modes of permeability creation result from frictional reactivation and/or brittle fracture of the crust and are typically accompanied by micro-earthquakes (MEQs).

The spatial distribution of the resulting microseismicity provides a method to monitor fracture development. MEQ data carry important information about the spatial distribution of hydraulic rock properties, such as permeability. This interpretation is based on the major hypothesis that MEQs result from a decrease in frictional resistance resulting from an increase in pore pressure thus triggering the reactivation of sliding along preexisting cracks. The generation of MEQs may signify the creation of porosity, with their locations hinting at the form and topology of the resultant architecture of connected permeable pathways. Where the crust is of sufficiently low initial permeability, the permeability may be increased by many orders of magnitude. This offers the prospect that changes in permeability may be defined if

[1]EMS Energy Institute, G3 Center and Department of Geosciences, Pennsylvania State University, University Park, USA. [2]EMS Energy Insititute, G3 Center and Department of Energy and Mineral Engineering, Pennsylvania State University, University Park, USA. [3]Department of Computer Science & Engineering, University of South Florida, Tampa, FL, USA. [4]Department of Computer Science and Engineering, Pennsylvania State University, University Park, PA, USA. [5]Pennsylvania State University, University Park, PA, USA. [6]Dipartimento di Scienze della Terra, La Sapienza Università di Roma, Roma, Italy. [7]Department of Engineering Science and Mechanics, Pennsylvania State University, University Park, PA, USA. ✉e-mail: pmy5077@psu.edu; elsworth@psu.edu

the energy release or other features of the MEQs are mechanistically linked to the creation of porosity and thereby permeability. Such a linkage requires a mechanistic connection between MEQs, fracture motion and changes in fracture morphology and wall rock damage[12,13].

Various methods and models have been proposed to estimate permeability based on seismicity and other related data. Effective permeability in a large rock volume may be evaluated from the estimation of hydraulic diffusivity consistent with the timing and location of the MEQ triggering front[14,15]. The hydraulic diffusivity is further applied by Chen et al. to image the final stage permeability distribution based on the tracer data with the occurrence time constraints of MEQs[16]. Continuous measurements of seismicity density may also be assimilated to estimate spatial permeability distribution based on Kalman filtering[17,18]. Earthquake hypocenters have been used as a proxy of pore pressure increase during well stimulation to invert the spatiotemporal permeability enhancement for Paralana EGS and Habanero EGS stimulations in Australia[19,20]. Relatively few studies have investigated the use of MEQ magnitude as a proxy for permeability or information regarding changes in fracture porosity. In a few cases, cumulative slip displacement (or cumulative seismic moment) has been used to estimate the fluid pressure distribution and from that infer the extent of the stimulated reservoir[21], and workflows accommodating MEQs in a more granular manner[12,13,22].

Illuminating the potential mechanistic relationship between MEQ characteristics and induced permeability changes requires access to high-quality datasets necessarily containing concurrent measurements of both quantities. These data should include accurate MEQ locations from a high-resolution seismic network together with local measurements of fluid injection pressures and volumes. Such high-quality data are rare but a number of field trials are now available where permeability has been purposely created through hydraulic stimulation in the subsurface with concurrent seismic measurements. These highly constrained field experiments offer the possibility to retrieve the form of the relation linking features of the MEQs to the observed change in permeability using Machine Learning (ML) methods – we utilize these rare datasets.

Due to their ability to identify obscured patterns and relationships, ML methods have recently been widely applied in the geosciences to extract broad patterns from large and noisy datasets[23-28]. Here, we process high-fidelity concurrent measurements of permeability changes driven by MEQs using machine learning (ML) models to discern linkages between injectivity (ratio of flow rate to injection pressure) and microseismicity and thereby constrain key underlying processes. Measured injectivities are first converted to permeability – a material property rather than an experimental response - then linked to the timing, location, and magnitude of MEQs. We use high-fidelity data from the EGS Collab and Utah FORGE 16 A (78)−32 well hydraulic stimulation studies - providing detailed concurrent time series of changes in permeability and MEQs - to constrain the functional relationship between seismicity and permeability using ML methods. We explore optimal formats for the ML strategy, confirm the form of key features linking the MEQs to permeability and define a functional relationship that is mechanistically consistent with this linkage. We use the ML system to make predictions on independent suites of data via transfer learning and demonstrate that key features of the seismicity-permeability dependency are universal.

## Results

We use high-fidelity injection pressure/rate and MEQ records from stimulation demonstrations at the EGS-Collab and Utah-FORGE projects where the objective is to create new porosity and hence permeability. We use these data to construct time history records of changes in permeability and connect those to moment magnitudes, locations and timing of MEQs. In particular, the discrete (in time) MEQ features are extracted and combined with the time-continuous pumping-derived permeability records and processed by ML models.

### EGS Collab hydraulic stimulation experiment datasets

Experiment 1 of the EGS-Collab project was one of a series of injection experiments to develop a mechanistic understanding of hydraulic stimulation in crystalline rock at decameter scale[29]. Separate injection and recovery holes are flanked by fans of monitoring holes to record rock mass displacements, electrical resistivity tomography (ERT) and seismic signals. This experiment is exceptionally well-constrained by the continuous monitoring and cataloging of active and passive seismic data (CASSM) throughout the injection period[30,31]. These meso-scale experiments were conducted at a depth of ~1.5 km, as accessed from an experimental adit, where depths and stresses are representative of real EGS reservoirs[32,33]. Five episodes of hydraulic stimulation were performed at EGS Collab in May 2018. The first two episodes (Ep1, and Ep2) used very low injection rates with few MEQs[29,34] and no permeability change signal to effectively utilize in constraining the MEQ-permeability relationship. However, we use data from the three subsequent continuous hydraulic stimulation episodes (Ep3, Ep4, and Ep5) where step-rate injections (Fig. 1a1, b1, c1) reactivated and created fractures adjacent to the injection borehole with significant signals for MEQs and permeability change. Injection episode Ep3 took place on May 24, 2018, using high injection rates to further propagate the fracture and allow it to connect to well E1-P, extending a nominal fracturing radius of 5.0 m generated in previous stimulation episodes[29]. The final two injection episodes (Ep4, Ep5) were conducted on May 25 with the goal of repeating the injection stimulation of Ep3 and to make additional measurements of the Step-rate Injection Method for Fracture In-situ Properties (SIMFIP)[29].

The location, time, and magnitudes of MEQs were recorded concurrently with time histories of injection[29,31]. In the hydraulic stimulations, seismicity initiated when pressure exceeded ~26 MPa (Fig. 1a2, b2, c2) suggesting that this stress level might represent the fracture propagation pressure[34,35]. Water jetting was observed by downhole camera in the recovery borehole during Ep5, indicating an hydraulic connection between injection (E1-I) and production (E1-P) wellbores from the remobilization or creation of fractures[29]. The spatiotemporal distribution of MEQs during the three episodes of stimulation approximate a radially expanding (cylindrical) geometry (Fig. 2)[13].

### Utah FORGE 16 A (78)−32 well hydraulic stimulation test datasets

The Utah Frontier Observatory for Research in Geothermal Energy (FORGE) is a field demonstration project testing the utility of multistage hydraulic stimulation for the development of Enhanced Geothermal Systems (EGS). Well 16 A (78)−32[36] is a deep (~2500 m) and highly deviated well initiating at the surface[37] with three separate stages of hydraulic stimulation conducted at three different locations near the toe of the well. Stimulation Stage 1 is in the open-hole wellbore at the toe of the well with Stages 2 and 3 being initiated through perforated intervals in the casing. Stage 1 was stimulated with water and Stages 2 and 3 with slickwater followed by crosslinked polymer, with Stage 3 also including the injection of proppant[38]. Similar to EGS Collab, a step-rate injection procedure was employed for all three hydraulic stimulation stages. Effective flowing networks or zones of permeability enhancement were generated during the three stimulations as evidenced by the signatures of relatively constant injection pressures (stage 1: Fig. 3a1) or decreasing injection pressure (stages 2 and 3: Fig. 3b1, c1) while increasing injection rate[36].

A high-resolution seismic network detected events as small as magnitude −2. A total of >2700 MEQs were recorded over the three-stage stimulation, with some continuing after the completion of each

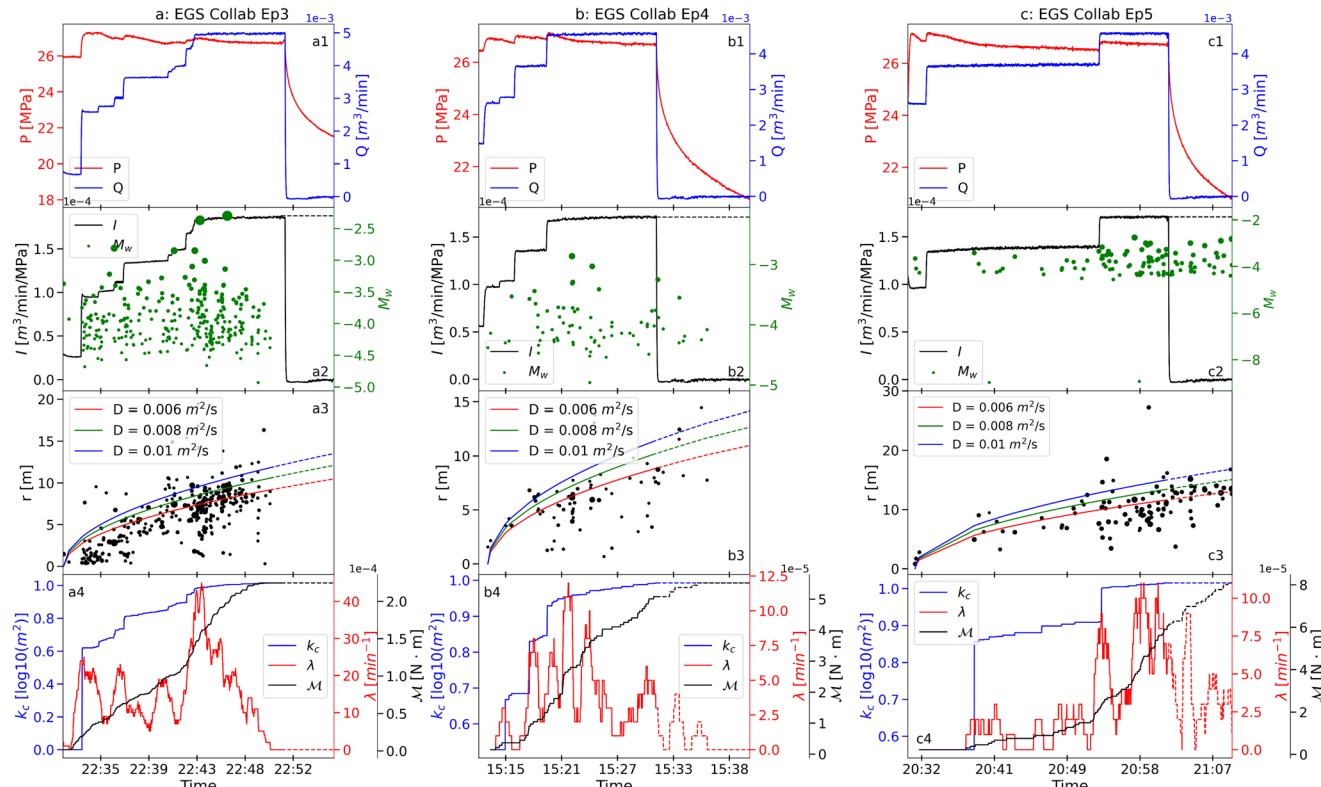

**Fig. 1 | Seismicity and injection observations for EGS Collab from Episode 3 (Ep3) to Episode 5 (Ep5).** The first row (**a1**–**c1**) shows the evolution of injection pressures (*P*) and injection rates (*Q*) during hydraulic stimulation. The second row (**a2**–**c2**) shows the time history of injectivity (*I*) and MEQ moment magnitudes (*M_w*).

The third row (**a3**–**c3**) shows the pressure-diffusive radius (*r*) fitted to the location of seismicity relative to the injection location (Shapiro et al., 1997; 2002). The fourth row (**a4**–**c4**) shows changes in permeability (*k_c*) and changes in two MEQ features (*viz.* seismicity rate (*λ*), and the cumulative log of seismic moment (*$\mathcal{M}$*)).

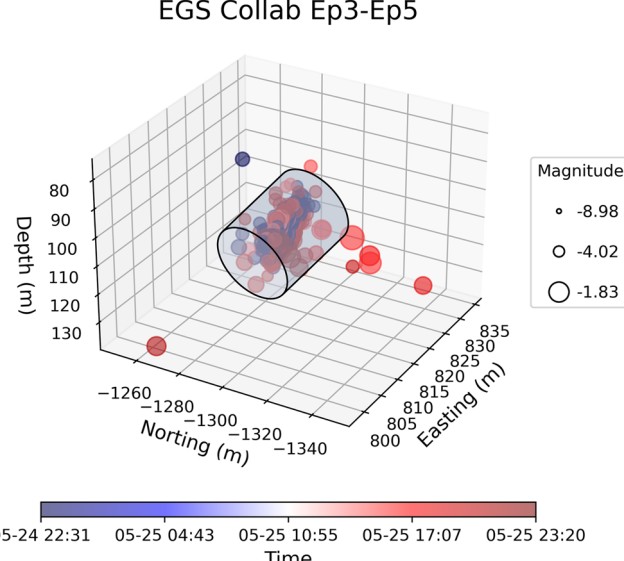

**Fig. 2 | Location, timing, and magnitude of MEQs recorded during EGS Collab stimulation tests Ep3-5, where event timing is shown by symbol color with magnitude scaled by symbol radius.** For the events shown, moment magnitude ranges from −8.98 to −1.83. Note radial migration of the seismicity.

stimulation stage. We filtered out the scattered MEQ data that were distant to the main MEQ cloud. We filtered out the scattered MEQ data that were distant to the main MEQ cloud; the spatiotemporal distribution of the filtered MEQs represent an approximately spherical zone, as shown in Fig.4.

## Measurement of permeability changes

Well injectivity, defined as the ratio of injection rate to wellhead pressure, is one useful proxy for monitoring the formation and evolution of fluid permeability[39] providing a simple diagnostic of the stimulation success. We evaluate injectivity for the full suite of injection data pertaining to the multiple episodes/stages in the two datasets. These data are punctuated by halts and shut-ins, as shown for EGS-Collab (Fig. 1a2–c2) and Utah FORGE (Fig. 3a2–c2). We cap recorded decreases in injectivity in the waning stages of injection at peak injectivity – as representative of the irreversible gain in permeability – since evolved permeability would not significantly decrease as excess pressures and flow rates drop to zero.

We use a diffusion model to follow the migration of the triggering front of the MEQ cloud[15,40,41]. For a homogeneous and isotropic medium, the triggering front is approximated as[15,40]:

$$r = \sqrt{4\pi D \Delta t} \tag{1}$$

where *r* is the separation between the migrating seismic front and injection point; *D* represents the best-fit hydraulic diffusivity; Δ*t* is the elapsed time since initiation of injection $t_0$, e.g, Δ$t = t_i - t_0$. A constant hydraulic diffusivity of $0.008 m^2/s$ is fitted for EGS Collab across all three episodes as shown in Fig. 1a3–c3 – representative of progressive stimulations of the same zone. Three different hydraulic diffusivity values (Fig. 3a3, b3, c3) are returned for the three stages at Utah FORGE, representative of the three different locations accessed along the wellbore.

We convert measured injectivities to mean permeability by defining approximated radial (EGS-Collab) or spherical (Utah FORGE) flow geometries representative of the different geometries of the

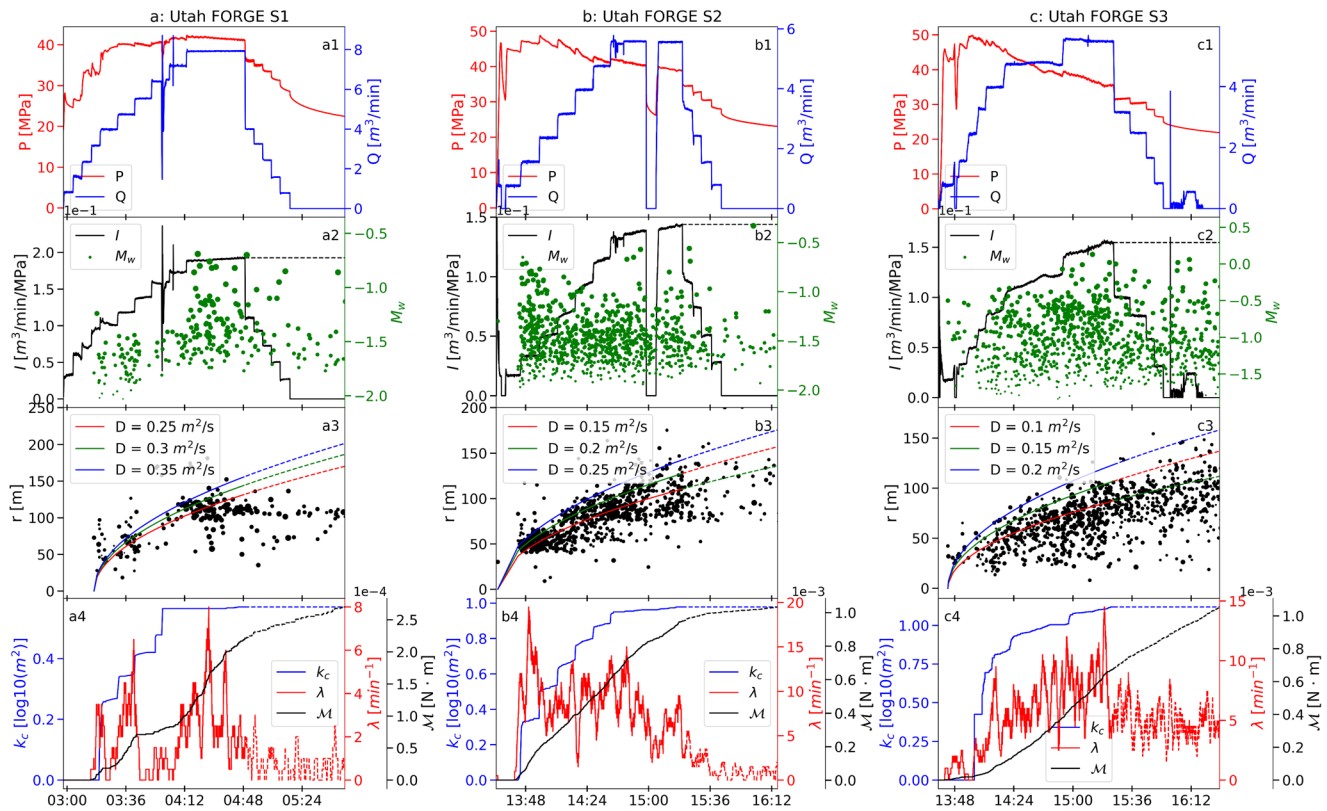

**Fig. 3 | Seismicity and injection observations for the three stages of hydraulic stimulation at Utah FORGE.** The first row (**a1**–**c1**) shows the evolution of injection pressure (*P*) and injection rate (*Q*) during hydraulic stimulation. The second row (**a2**–**c2**) shows the time history of injectivity (*I*) and MEQ moment magnitudes (*M_w*).

The third row (**a3**–**c3**) shows the pressure-diffusive radius (*r*) fitted to the location of seismicity relative to the injection location (Shapiro et al., 1997; 2002). The fourth row (**a4**–**c4**) shows permeability changes (*k_c*) and changes in two MEQ features (*viz.* seismicity rate (*λ*), and cumulative logarithm of seismic moment (*M*)).

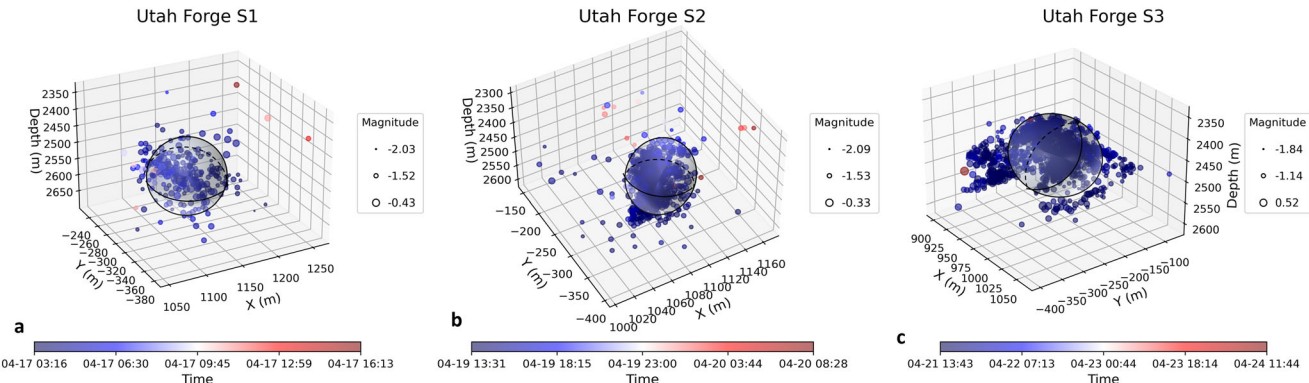

**Fig. 4 | Characteristics of microseismic events in Utah FORGE stimulation tests.** **a**–**c** show the location, timing, and magnitude of MEQs across Stages 1 to 3 (S1–S3). The event timing is shown by symbol color with magnitude scaled by symbol radius. The moment magnitude range of the events spans from −2.09 to 0.52. Note spherical migration of the seismicity.

evolving seismicity clouds and wellbores – and assuming steady flow within the evolving cloud. For EGS Collab, the seismic front (Fig. 2) propagates radially from the injection wellbore that is long in relation to the distal radius of the seismicity front, ultimately representing the external far-field pressure boundary. For FORGE, the injection zone length is short in comparison to the distal pressure boundary (Fig. 4) and evocative of spherical propagation of the seismicity front and corresponding far-field pressure boundary. The permeability for the EGS Collab experiment is thus:

$$k = \frac{\mu I}{2\pi h} \ln\left(\frac{r_t}{r_w}\right) \qquad (2)$$

where *k* is the average permeability, *μ* is the viscosity of the injected fluid (water) accounting for borehole temperature, *h* is borehole/cylindrical-zone length and *I* is the injectivity defined as the ratio of flow rate (*Q*) to pressure differential ($\Delta P = P_d - P_e$). $P_d$ is the downhole pressure, $P_e$ is the pressure at the pressure external boundary, coincident with the seismicity front. The parameter $r_w$ is the interior injection wellbore radius and $r_t$ is the radius to the external flow boundary observed and utilized for Eq.(1). Similarly, for Utah FORGE, the average permeability is expressed as:

$$k = \frac{\mu I}{4\pi}\left(\frac{1}{r_w} - \frac{1}{r_t}\right) \qquad (3)$$

where radii represent the spherical geometry of the stimulated zone. We note that it is always difficult to accurately characterize the MEQ distribution using exact geometry. Here we assumed a simple spherical MEQ distribution as broadly characteristic of the flow geometry based on the MEQ observations. This assumption as spherical (Utah FORGE) or radial (EGS Collab) flow does not affect the process of linking MEQs to permeability changes since permeabilities calculated from the two geometries are linked to injectivity through a constant representing the flow geometry.

Since initial permeabilities are low ($\sim 10^{-17} m^2$) and we are interested in the observed change from those low initial values. Therefore, we introduce normalized permeability change $k_c$, $k_c = \log k/k_0$, defined as the ratio of permeability at a given time to the initial reservoir permeability and use this to characterize permeability evolution, with positive $k_c$ indicating permeability enhancement. Being a dimensionless parameter, $k_c$ also aids direct comparison between different sites and stimulation experiments. Figures 1a4, b4, c4, 3a4, b4, c4 illustrate the normalized permeability changes during the entire stimulation period for EGS Collab Ep3-Ep5 and Utah FORGE S1-S3, respectively. The data show that permeability increases monotonically as constrained by cropping all terminal declines in injectivity as the injection wanes. Again, this is consistent with the notion that most of the permeability enhancement is irreversible. We note that the best-fit hydraulic diffusivity used in this study might not be the only one that matches the migration of the triggering front for the MEQs data. However, it would not affect the calculation of permeability changes, as the effect of hydraulic diffusivity is factored out; instead, the location of the migrating pressure front used as a proxy for the evolving steady flow geometry.

## MEQ feature extraction

We extract features from the available MEQ catalogs spanning the stimulation injection period and attempt to link these to the observed changes in permeability. We exclude MEQs recorded post-stimulation in this study, when events could have been triggered either by the diffusive expansion of fluid pressure decay[42] or poroelastic stressing[43], as these likely do not significantly alter permeability. Two features were extracted from the MEQs catalog: seismicity rate ($\lambda$) and cumulative logarithmic seismic moment ($\mathcal{M}$). As the MEQ locations were used to define the evolving flow radius and to then calculate permeability, we did not extract features related to MEQ locations to avoid the risk of artificially correlating MEQs to permeability changes.

Higher injection rates or cumulative hydraulic energy (the time integral of the product of wellhead pressure and injection rate) are associated with increased seismicity rates, which might provide insight into reservoir connectedness[27]. Also, direct observations from both EGS Collab (e.g., Fig. 1a2) and Utah FORGE (e.g., Fig. 3b2) indicate that injectivity changes are associated with changes in seismicity rate, suggesting an underlying association between seismicity rate and permeability change. The seismicity rate, $\lambda_i$, at injection time, $t_i$, is calculated by summing the number of events in the following interval $[t_i, t_i + \Delta t_w]$, and dividing by the interval length, $\Delta t_w$. This averaging approach, using a backward-looking moving time window of duration $\Delta t_w$, imposes a controllable degree of smoothing on the stochastic earthquake process (Supplementary Fig. S4, Fig.S5). In this study, $\Delta t_w$ was set as 2 min and the seismicity rate changes over time for both EGS Collab and Utah FORGE are depicted in Figs. 1a4−c4, 3a4−c4, respectively. The rationale and sensitivity analysis of $\Delta t_w$ is explored and discussed in next section.

The second feature extracted is the cumulative logarithmic seismic moment. This metric has been instrumental in estimating the size of the activated reservoir volume, leading to the strategic placement of a new production well[21,44]. Moreover, a variety of field studies on seismicity triggered by fluid injection have shown that the total release of seismic energy (or seismic moment) is directly correlated with

hydraulic energy[45,46]. The cumulative logarithmic seismic moment, $\mathcal{M}$, is defined as the cumulative sum of seismic moment during the cumulative time interval of $[0, t_i + \Delta t_w]$, expressed as:

$$\mathcal{M} = \sum_{t=0}^{t_i + \Delta t_w} \log M_0 \tag{4}$$

Here, $M_0$ is the seismic moment, converted from the moment magnitude, $M_w$ as[47]:

$$\log M_0 = 1.5 M_w + 13.5 \tag{5}$$

Using seismic moment $M_0$, rather than moment magnitude $M_w$, provides a direct connection to rupture area and slip; it also avoids the issue of accommodating negative moment magnitudes while representing an integration that directly scales with strain energy release. The evolution of cumulative logarithmic seismic moment ($\mathcal{M}$) with time is illustrated in Figs. 1a4−c4, 3a4−c4 for three episodes of the EGS Collab and three stages of the Utah FORGE hydraulic stimulation experiments.

## Stand-alone models for EGS Collab and Utah FORGE datasets

We use our observations of MEQs and local permeability creation to define a framework for predicting permeability evolution utilizing a Bidirectional Long Short Term Memory (Bi-LSTM) stateful neural network (See Methods section)[48,49]. We note for our observations that stateful models such as LSTM are well suited for modelling sequential data and capturing the temporal dependence than stateless neural networks[50]. In 'uni-directional' LSTM models, the state at a given time captures the data history i.e., information in the preceding data samples. A bi-directional LSTM model is advantageous when the output (permeability change) depends on the entire predictor sequence (seismic moment) as it captures both backward and forward dependencies through time[50]. Additionally, bi-directional LSTM models provide improved stability and faster convergence, as detailed by analysis in the Supplementary Table S11 and Text. The goal is to forecast permeability evolution (model output) from features of the MEQs (input features), namely seismicity rate, $\lambda_i$, and cumulative logarithmic seismic moment, $\mathcal{M}_i$ extracted from the EGS Collab and Utah FORGE datasets. We evaluate the models for each dataset and then the generalizability of the model across these two datasets through transfer learning.

We test two deep learning models - Bidirectional Long-short Term Memory (Bi-LSTM) and Bidirectional Gated Recurrent Unit (Bi-GRU) - and observe stable performance using the Bi-LSTM model (Supplementary Table S1 and Table S2). Our best-performing Bi-LSTM model for the EGS Collab dataset consists of 2 hidden layers with 128 nodes each and one linear layer. A batch size of 105, a learning rate of 0.0003 and 700 epochs are utilized. We employ the modified physics-informed loss function introduced earlier (Eq. 8) together with the Adam optimizer[51] and the $R^2$ score metric to evaluate model performance. As shown in Fig. 5a, the trained EGS Collab Bi-LSTM model replicates permeability evolution over time with remarkable accuracy – as demonstrated by the excellent test $R^2$ score of 0.937. The monotonically increasing prediction of permeability attests to the effectiveness of our physics-constrained loss function. Notably, the Bi-LSTM EGS Collab model accurately captures the ultimate permeability value, although it may not fully match the intermediate time history.

For the Utah FORGE dataset, we adopt a similar Bi-LSTM structure to predict permeability changes based on MEQ features. Our best-performing model for this dataset has two hidden layers with 64 nodes each, a batch size of 96, a learning rate of 0.001 with 150 epochs. As illustrated in Fig. 5b3, the Utah FORGE Bi-LSTM model also returns high fidelity fits across the training, validation and test sets. Although the test $R^2$ score is 0.85, our predicted terminal permeability aligns well

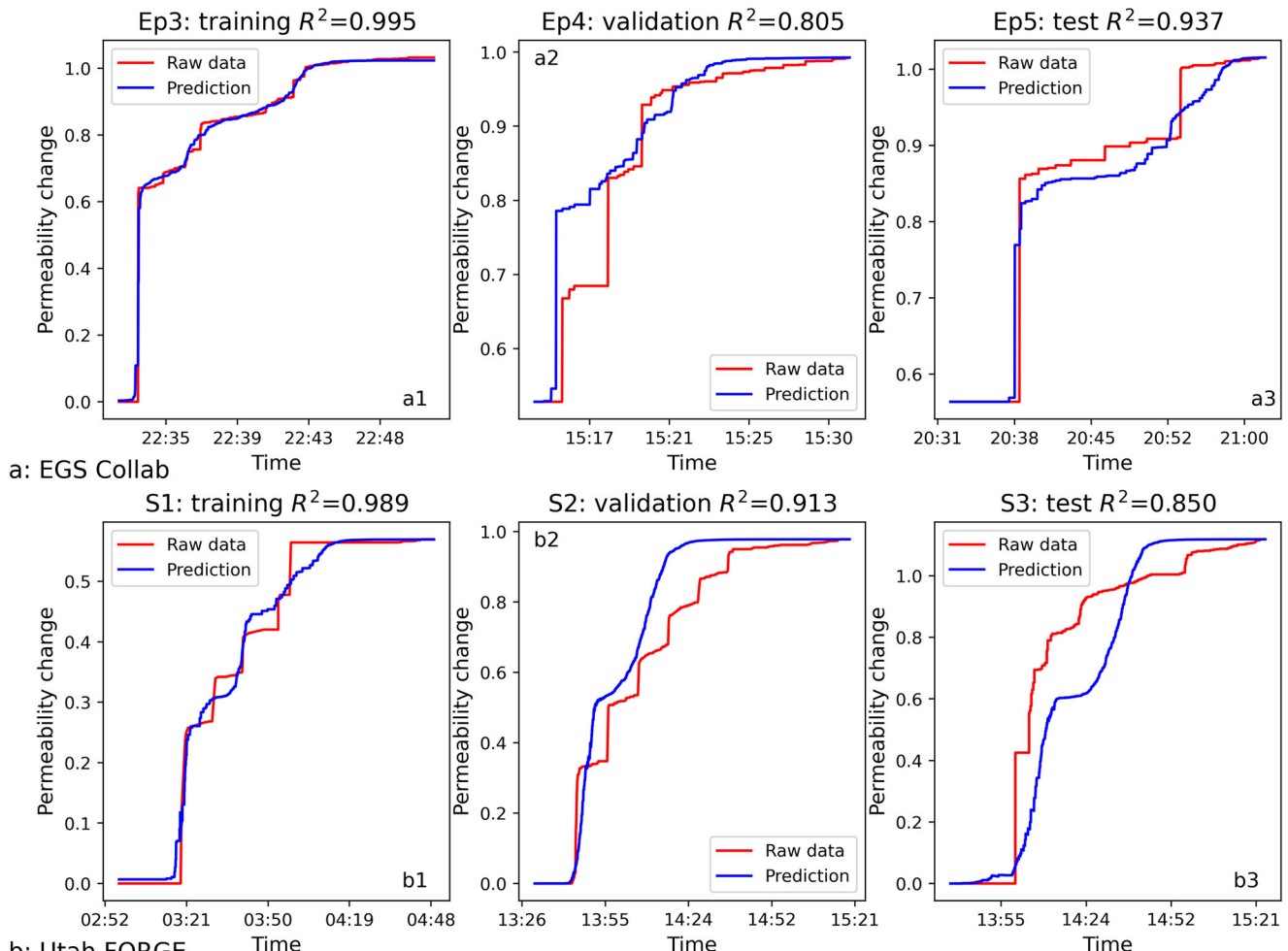

**Fig. 5 | Comparison between raw permeability data (ground truth) and predictions from training, validation and test sets for EGS-Collab and Utah FORGE datasets, respectively.** The first row shows the EGS Collab Bi-LSTM model outputs compared with raw data using Ep3 for training (**a1**), Ep4 for validation (**a2**) and Ep5 for testing (**a3**). The second row shows the Utah FORGE Bi-LSTM model predictions compared with raw data using Stage 1 for training (**b1**), Stage 2 for validation (**b2**) and Stage 3 for testing (**b3**).

with the observed value. One possible explanation for the relatively poorer performance on the test set (Stage 3) using the model trained on Stage 1 may lie in the different hydraulic stimulation operations employed between these two stages. Specifically, Stage 1 was conducted on the open-hole wellbore section, whereas Stage 3 targeted a perforated interval. Additionally, variations in the types of working fluids used across these stages[36] could contribute to distinct patterns in permeability evolution. Second, from a machine learning theoretical perspective, the discrepancy in the performance between Bi-LSTM and Bi-GRU lies in their convergence and stability characteristics (Supplementary Table S1 and Table S2)[52,53]. If the model fails to reach a stable point, then its prediction will be inconsistent, hampering its generalization performance. In this study, we adapt a grid search to improve performance; although one can find better model parameters for this problem by using neural architecture search that will lead to optimal performance. For the parameter $\Delta t_w$ used for calculating seismicity rate feature, we studied the model performance under different $\Delta t_w$ for both EGS Collab and Utah FORGE datasets. As shown in Supplementary Tables S9, S10, the Bi-LSTM model with $\Delta t_w = 2$ min returns the best $R^2$ scores on both validation and test datasets although a range of values all return acceptable results. An optimal result may exist since a short $\Delta t_w$ misses the resulting change in permeability, consistent with the expectation that hydraulic response time is finite, and a long $\Delta t_w$ smooths and smears the seismic response and

effectively removes information from the data in the intensive hydraulic stimulation processes (Supplementary Figs. S4, S5). Interesting, the $R^2$ does not change significantly with increasing of $\Delta t_w$ (when $\Delta t_w > 2$ min) especially for the Utah FORGE dataset as shown in Supplementary Table S10. This insensitivity may reflect the fact that cumulative logarithmic seismic moment is the key feature in predicting permeability changes, and this feature does not change under different $\Delta t_w$.

## Transfer learning

We use transfer learning to evaluate the generalizability of the models across the two datasets. Specifically, the trained Utah FORGE Bi-LSTM model is used to make predictions for the EGS Collab data, then vice versa.

The transfer learned model for the EGS Collab dataset is based on the stand-alone Utah FORGE Bi-LSTM model (reference model). This is accomplished by using the same Bi-LSTM architecture as the reference model and using the pre-trained weights (trained and validated on the Utah FORGE dataset) for model initialization. Next, the model weights and hyperparameters are fine-tuned using Episode 3 of EGS Collab resulting in a learning rate of $1 \times 10^{-3}$, a batch size of 104 with 650 epochs for the transfer learned model. Similarly, a transfer learned model for Utah FORGE dataset was constructed based on the stand-alone EGS Collab Bi-LSTM. The Utah FORGE transfer learned model

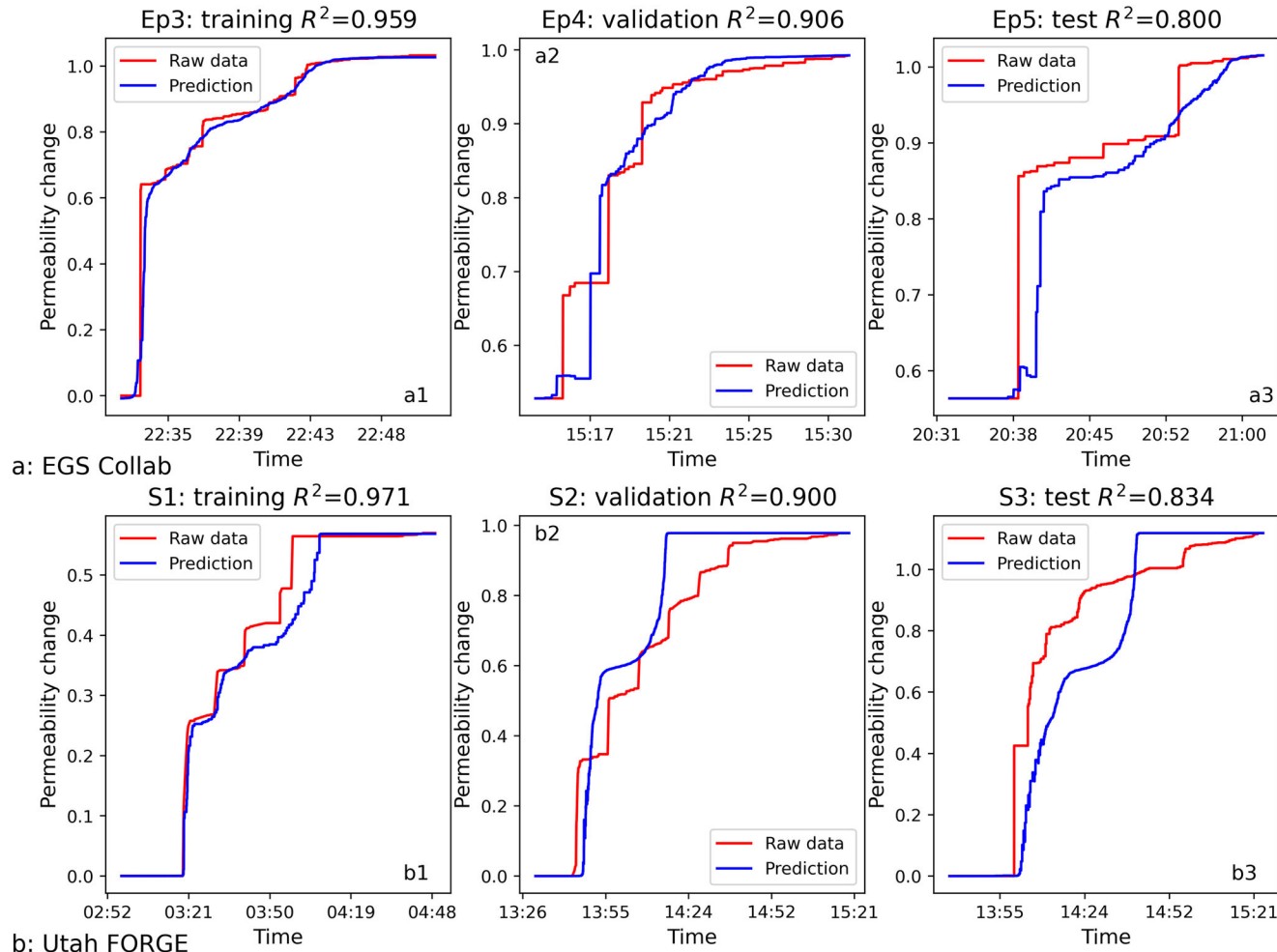

**Fig. 6 | Comparison between ground truth and transfer learning prediction results for EGS-Collab and Utah FORGE datasets, respectively.** The first row shows the results of transfer learning applied to the EGS Collab dataset results comparison (Ep3 for training (**a1**), Ep4 for validation (**a2**), Ep5 for testing (**a3**)) using the Utah FORGE Bi-LSTM model. The second row shows the results of transfer learning applied to the Utah FORGE dataset results comparison (Stage 1 for training (**b1**), Stage 2 for validation (**b2**), Stage 3 for testing (**b3**)) using the EGS Collab Bi-LSTM model. Note that the predictions are quite good for both cases of transfer learning.

was trained using 110 epochs, a batch size of 96 and a learning rate of $1 \times 10^{-3}$. The Adam optimization algorithm and adjusted physics-constrained loss function were employed for all transfer learning models. Detailed evaluation of transfer learning capability for various deep learning models across two datasets is demonstrated in Supplementary Tables S5, S6.

As depicted in Fig. 6a, the EGS Collab transfer learned model yielded accurate predictions of permeability on both validation (Episode 4) and test (Episode 5) sets, with $R^2$ scores of 0.91 and 0.80, respectively. Likewise, it was possible to successfully transfer the EGS Collab Bi-LSTM model to the Utah FORGE dataset, as illustrated in Fig. 6b. Moreover, both transfer learned models were capable of predicting the ultimate permeability of hydraulic stimulations. The success of transfer learning underscore the domain-independence and generalizability of the extracted MEQ features in predicting permeability evolution – and hint to the prospect that a robust causative physical linkage may exist between the creation of porosity/permeability and release of strain energy indexed by cumulative MEQ magnitudes. Nevertheless, prediction accuracy of the model may be constrained by the quality of the dataset. This is particularly true for datasets that necessitate concurrent measurements of accurate MEQ locations from high-resolution seismic networks, along with local measurements of fluid injection pressures and volumes. For

a relatively small dataset or when the relationship is simple, and all dataset splits come from the same distribution, one could observe machine learning models performing comparably to deep learning models; however, in the scenarios involving out-of-distribution and transfer learning, neural-based models often have an advantage over classical ML models[54-56].

## Discussion

The good performance and transferability of our ML models suggest that changes in permeability can be associated with particular MEQ features, namely seismicity rate and cumulative logarithmic seismic moment. In particular, the successful implementation of transfer learning across the two independent datasets suggests a generalization of this association. Thus, with a connection tentatively established, a question remains as to the nature, underlying physics and scaling of such relationships. Seismicity rate can be transformed into seismic moment based on the Gutenberg-Richter magnitude-frequency relationship[57] with the cumulative logarithmic seismic moment integrating both the frequency and magnitude information of MEQs – akin to scaling with cumulative energy release. Therefore, functional relationships linking permeability change and seismic moment appear physically motivated and plausible. EGS reservoir stress state and material properties may be quantitatively correlated with MEQ

magnitudes by noting that for an observed limited range of stress drops, seismic moment correlates with fault slip area. Thus, for this slipping patch, a fractal scaling of roughness will condition a larger dilation for a larger patch size – and hence a larger change in permeability[22]. Furthering this logic, we explore physical connections between seismic moment of MEQs and permeability evolution.

Changes in the permeability of individual fractures may be related to the change in aperture $\Delta b$ from an original aperture $b_0$ as:

$$\Delta k = \frac{(b_0 + \Delta b)^2}{12} - \frac{b_0^2}{12} \qquad (6)$$

and to overall bulk permeability or transmissibility via the cubic law as[58]:

$$\Delta k = \frac{(b_0 + \Delta b)^3}{12s} - \frac{b_0^3}{12s} \qquad (7)$$

where $s$ is the spacing between adjacent parallel fractures. Where initial permeability is small in comparison to the change in permeability, then $b_0 \ll \Delta b$ and Eq. (10) reduces to $\Delta k \sim \frac{\Delta b^3}{12s}$. Where fractures remobilize in shear, then aperture change is controlled by dilation over the existing fracture topography – the amplitude of which grows with the dimension of the mobilized patch. For failure in shear, the aperture will increase by an increment $b_s$ as conditioned by a slip offset $\Delta u_s$ and fracture dilation angle $i$, as[59]:

$$b_s = \Delta u_s \tan i \qquad (8)$$

Similarly, seismic moment, $M_0$, is defined as:

$$M_0 = M_0^s = GA\Delta u_s \qquad (9)$$

where $M_0^s$ is the seismic energy released for slip on a fault patch of area $A$ embedded within a medium of average shear modulus $G$. For fracture extension by tensile opening, this equivalent moment, $M_0$ is defined as[60]:

$$M_0 = M_0^n = 2GA\Delta u_n \qquad (10)$$

where the $M_0^n$ is the seismic energy released during fracture opening or closing and $\Delta u_n$ is the normal displacement in crack opening or closing.

Incorporating Eqs. (10–12), the seismic moment in shear $M_0^s$ is linked to change in permeability $\Delta k$ as:

$$M_0^s = \frac{GA(12s\Delta k_s)^{1/3}}{\tan i} \qquad (11)$$

where $\Delta k_s$ is the permeability change resulting from fracture shearing. Similarly, using Eq. (10) and Eq. (13), the moment in tensile opening, $M_0^n$ may be expressed as:

$$M_0^n = 2GA(12s\Delta k_n)^{1/3} \qquad (12)$$

where $\Delta k_n$ is the permeability change resulting from fracture opening in extension. In addition, the seismic moment released from a volume $V$ surrounding a fault may also be expressed as[57]:

$$M_0 = V\Delta\tau \qquad (13)$$

where $\Delta\tau$ is the shear stress drop, typically in the range 0.1-10MPa[61–63]. Assuming this volume scales with the area of the transected fault of edge dimension $a$ [63] then for a prismatic fault area $A \sim a^2$ and volume

$V \sim a^3$ enabling the scaling $A \sim V^{2/3}$ to be established. Thus, incorporating the scaling between moment $M_0$ and fault area, $A$, an analogous destressed volume $V$ may be defined by combining Eq. (14) and Eq. (16) with $A \sim V^{2/3}$ to link permeability change directly with seismic moment in shear $M_0^s$ as:

$$\Delta k = \Delta k_s = M_0^s \Delta\tau^2 \tan^3 i/12sG^3 \qquad (14)$$

Similarly, combining Eqs. (15) and (16) defines permeability change due to tensile opening as:

$$\Delta k = \Delta k_n = M_0^n \Delta\tau^2/96sG^3 \qquad (15)$$

Thus, where permeability is generated by MEQs in shear, tension or in mixed modes, Eqs. (17) and (18) reflect the surprising linear proportionality between permeability change and seismic moment in either shear or tension as: $\Delta k_s \propto M_0^s$ and $\Delta k_n \propto M_0^n$. Shear modulus and stress drop vary within narrow bounds[63], and fracture spacing for a given location with reactivating pre-existing fractures will be invariant.

Moreover, extensive earthquake observations indicate a proportionality between seismic moment and slip area[63]:

$$\log A \propto \frac{2}{3} \log M_0 \qquad (16)$$

albeit for larger earthquakes than those considered here. This observed scaling is a manifestation of Eq. (16) with the proportionality $A \propto V^{2/3}$. Thus, Eq. (19), may also be used to confirm the scaling relationship linking changes in permeability and seismic moment. By substituting Eq. (19) into Eq. (14) or Eq. (15), a change in permeability is proportional to the respective seismic moments as $\Delta k \propto M_0$ reaffirming the relations of Eqs. 17, 18. Since individually in both shear and tension $\Delta k \propto M_0$ then permeability enhancement resulting from mixed mode failure would logically also conform to this scaling. Hence the quality in the predictive capability and transferability of the Bi-LSTM models. We also test standard machine learning models such as linear regression, XGBoost, elastic-net and ensemble methods (Voting Regression) for our standalone experiments in both the EGS Collab and Utah FORGE datasets (Supplementary Tables S1, S2). For our Voting Regression ensemble model, we combine decision-trees, XGBoost and Elastic-net models, confirming the linear relationship among the features and output variables, aligning with the posited linear theoretical relationship between permeability changes and seismic moment.

Of course, if the scaling $\Delta k \propto M_0$ holds, then a linear relation should be apparent in the incremental form $\Delta k \propto M_0$ for individual events or in the time integrated form as $\int \Delta k \propto \int M_0$ where the integration is for successive events as $k/k_0 \propto \Sigma M_0$. However, plots of normalized permeability change ($\Delta k/k_0$) versus seismic moment ($M_0$) and normalized permeability ($k/k_0$) versus cumulative seismic moment ($\sum M_0$) do not reflect this (Supplementary Figs. S2, S3). This mismatch may be explained by considering adherence to the fundamental assumption that incremental changes in permeability ($\Delta k_n$) are larger than initial ($k_0$) or evolving permeability ($k_n$) as $\Delta k_n \gg k_n$. This is likely true for the first MEQ, but not for the last.

For the first MEQ: $k_0 \to 0$ therefore $b_0 \to 0$ thus $\Delta k \propto (b_0 + \Delta b)^3 \propto \Delta b^3$ and the scaling relations $\Delta k \propto M_0$ and $\int \Delta k \propto \int M_0$ as equivalent to $k/k_0 \propto \Sigma M_0$, as derived previously, should all hold. Thus, the early time evolution of permeability with time (Figs. 6, 7) and with cumulative seismic moment (Supplementary Figs. S2, S3) should evolve with an initial but unprescribed gradient, as apparent in the steep gradients in these figures.

For the last MEQ: $\Delta k_n \ll k_n$ therefore $\Delta b_n \ll b_n$ thus $\Delta k_n \propto (b_n + \Delta b_n)^3 \propto b_n^3$ and aperture and therefore permeability will change little with successive MEQs or with time. Thus, the time history of

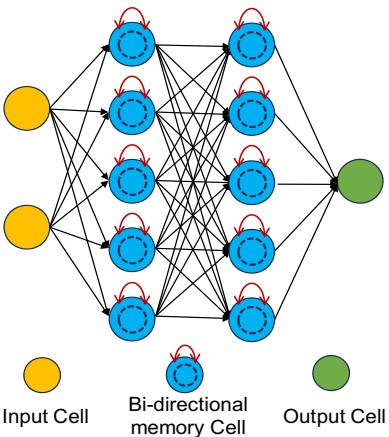

**Fig. 7 | Schematic representation of the Bi-LSTM neural network used in this study.** The Bi-LSTM structure includes input cell (yellow), bi-directional memory cell (blue), and output cell (green). The bi-directional memory cells are capable of processing data sequences in both forward and backward directions, denoted by the red arrows, to capture temporal dependencies for improved prediction accuracy.

permeability at late time will asymptote to zero gradient – the form apparent in Figs. 6, 7 and in the permeability versus cumulative seismic moment of Supplementary Figs. S2, S3. This concave downwards evolution of these plots represents the change from $\Delta k \propto M_0$ and $k/k_0 \propto \Sigma M_0$ scaling being appropriate for the first MEQ-driven change in permeability, but not for the last or late term changes where the assumption of preceding permeability $k_n \to 0$ is no longer valid at late time.

However, this early-time linear scaling between permeability change and seismic moment in part explains the robustness of the predictions recovered from the Bi-LSTM models applied individually as well as the strong performance of the two transfer learned models – even though the late-time response departs from this scaling. The training of the ML routines is able to accommodate this evolution and hence provide robust estimates of evolving permeability. These results guide our understanding of permeability evolution within fractured reservoirs where MEQs are ubiquitous and implicated in the evolution of permeability while offering a framework to predict and control stimulation outcomes regardless of failure mode. Such an ability to predict changes in permeability and to adapt stimulation methods for desired outcomes can lead to optimized stimulation techniques improving the efficiency and sustainability of subsurface recovery of energy and fuels and promoting the understanding of natural evolution of crustal permeability. It is important to note that the relationship between injection volume and cumulative seismic moment may not always hold in the case of larger earthquakes induced by fault reactivation, such as the Mw-5.5 event during the Pohang EGS stimulation[64,65] and several events larger than 5.0 in the Sichuan Basin during shale gas fracturing[66]. In our study, we focus on linking small MEQs to permeability enhancement during hydraulic stimulations. The largest moment magnitudes of events at the EGS Collab and Utah FORGE are −1.83 (Fig. 2) and 0.52 (Fig. 4), respectively. These small MEQs result from the creation of porosity, suggesting the form and topology of the resultant architecture of connected permeable pathways. Larger MEQs would also impact permeability in a similar way but large-scale reactivation of a discrete fault is not apparent in these field studies.

We have explored the role of microearthquakes (MEQs) during hydraulic stimulation in driving permeability changes. The EGS Collab and Utah FORGE datasets where high resolution and concurrent measurements of permeability and seismicity (MEQs) are available allow meaningful constraint of these processes. Observations of raw injection flow rate data constrained by injection pressures allow

injectivity to be causally linked to MEQs and in turn to changes in permeability. Select features of the MEQs are used to develop machine learning (ML) models to predict observed changes in injectivity processed to define the corresponding changes in permeability. This raw dataset used is then linked using machine learning (ML) to develop relationships between permeability changes and key features for the EGS Collab and Utah FORGE datasets. For both datasets, Bi-LSTM models constrained by a basic physics-based constraint are trained, validated then tested to predict permeability evolution using the MEQ features namely, the seismicity rate and the cumulative logarithmic seismic moment. Success of transfer learning confirms the generalizability of the models i.e., the potentially universal connection between the extracted MEQ features and permeability change thus suggesting that response of distinct sites and stimulation scenarios may be predictable from data from other disparate geographic regions and geologic terranes with little additional model fine tuning.

Leveraging sample-by-sampling normalization and adjusted loss functions, our best-performing stand-alone Bi-LSTM models have demonstrated promising results in predicting permeability changes over time for both EGS Collab and Utah FORGE datasets. The sample-by-sample normalization addresses the challenge associated with the different variable range in training vs validation and test sets. The model performances on validation and test dataset reflect our physical understanding of the differences across stimulation episodes and stages. The transfer learned models showcase the generalizability of the models across the two datasets, alluding to an underlying physical connection.

These insights lay the groundwork for a deeper understanding of the interactions between microseismic events and the evolution of reservoir permeability. Importantly, theoretical arguments linking anticipated changes in fracture network permeability with the seismic moment release from the MEQs suggest a linear linkage as $\Delta k \propto M_0$ and *a posteriori* infer the success of the prediction for the ML models. The physical interpretations of our functional equations elucidate the underlying mechanics of hydraulic stimulation processes, confirming the key features linking with permeability obtained from ML models. Although complexities and uncertainties present challenges, the established relationships form a valuable theoretical foundation for further research and practical applications in the context of optimized stimulation techniques, efficient energy extraction, seismic mitigation for EGS reservoir and unconventional hydrocarbon recovery and for understanding the evolution of permeability in the crust.

## Methods
EGS Collab Episodes 3, 4, and 5 are discrete experiments conducted sequentially at the same location[29]. To preserve the sequence, Episode 3 was used as the training set and Episode 4 as the validation set to perform hyper-parameter optimization for the Bi-LSTM models. Episode 5 served as the unseen test set to evaluate the generalization capability of the trained model. Similarly, for the Utah FORGE dataset, Stage 1 serves for training and Stage 2 for validation, with Stage 3 reserved for testing.

### Normalization
Min-max normalization or standardization based on the training set statistics constitutes the first pre-processing step. Initial data analysis for both EGS Collab and Utah FORGE stimulations revealed different distributions and data ranges for seismicity rates and moments across all episodes/stages (Supplementary Fig. 1). For example, the seismicity rate, $\lambda$, ranged between 0 and 45 events per minute in EGS Collab Episode 3 (Fig. 1a4), but only 0-12 and 0-11 for Episode 4 (Fig. 1b4) and Episode 5 (Fig. 1c4), respectively. Similarly, for Utah FORGE, the Stage 1 cumulative logarithmic seismic moment, $M$, increased from 0 to $2.5 \times 10^3$ (Fig. 3a4), to a much broader range of 0 to $1 \times 10^4$ in Stage 2 (Fig. 3b4) and Stage 3 (Fig.3c4). However, the need to preserve

the time series nature of the data precludes the common practice of combination and shuffling of all data at different stages/episodes into a unified dataset for subsequent training, validation and test set splitting. As a result, training, validation, and test sets (i.e., distinct episodes or stages) have very different statistics making it impossible to implement the traditional normalization or scandalization approaches. To address this challenge, we used a novel normalization method for sample-by-sample min-max normalization for the validation and test sets, while a traditional min-max normalization method was applied to the training set.

Data can only be normalized using the statistics of seen data i.e., the training set. The essence of the sample-by-sample min-max normalization method is to perform the normalization based on the assumption that samples are introduced in a pure online learning scenario, where the model is exposed to single sample at a time. This introduces a mechanism of calculating a running min and max over the unseen distribution, without any need of knowing the entire distributions statistics. For the normalization of the $i_{th}$ sample of the validation set, we first combine the training set and all *seen* samples up to the $i_{th}$ sample normalization as dataset $X^{val}$, expressed as:

$$X^{val} = [x_1^{train}, x_2^{train}, \ldots, x_n^{train}, x_1^{val}, x_2^{val}, \ldots, x_i^{val}] \qquad (17)$$

where $x_j^{train}(j=1,2,\ldots,n)$ represents a sample in the training set, $n$ is the length of the training set and $x_j^{val}(j=1,2,\ldots,i)$ is a sample in the *seen* validation set. Then, we perform the min-max normalization for the $i^{th}$ sample ($x_i^{val}$) in the validation dataset by finding the minimum and maximum values of combined dataset $X^{val}$, e.g.:

$$x_i^{valnorm} = \frac{x_i^{val} - \min(X^{val})}{\max(X^{val}) - \min(X^{val})} \qquad (18)$$

The same process is applied to each sample in the test set. Here, we only use all *seen* samples in the test set to combine with the training and validation sets. This approach resolves the issue of range disparity between training, validation, and testing sets without violating the normalization principles.

### Bi-LSTM network structure
The basic Bi-LSTM network structure used in this study is depicted in Fig. 7, including input cells, output cells, and Bi-directional memory cells. The Bi-directional memory cells incorporate both backward and forward information about the sequence at every time step. This processes the input cells with two hidden states, enabling them to preserve information from both past and future at any point in time. For details of the internal structure of the Bi-directional memory cell, such as the forget gate, update gate, and output gate, we refer readers to Sepp & Jurgen[49].

### Physical informed loss function
To preserve the monotonically increasing trend of permeability during the stimulation process, we introduce a new loss function which extends the traditional mean squared error (MSE) objective function with an additional penalty term which helps stabilize the learning by providing improved gradient estimates. The adjusted loss function, $\mathcal{L}$, is given by:

$$\mathcal{L} = MSE + \alpha \times |y(t_i) - y(t_{i-1})| \qquad (19)$$

Here, $y(t_i)$ and $y(t_{i-1})$ are the prediction output values at time $t_i$ and $t_{i-1}$, respectively. The parameter $\alpha$ is a non-negative scalar penalty coefficient, which is set to 0 when $y(t_i) \geq y(t_{i-1})$; otherwise, $\alpha$ is set to $|\alpha|$ to ensure that it is positive. We used $\alpha$ values of 1000 and 150 for the EGS Collab and Utah FORGE datasets, respectively. We compared model performance between models using a proposed physically

informed loss function and using a standard MES loss function ($\alpha = 0$) with results shown in Supplementary Tables S7, S8 – these indicate that adding the physics-based constraint to the loss function aids convergence and better represents the relationship among different variables compared to standard regression loss. We also compared model performance across various $\alpha$ values (Supplementary Fig. S6, S7). As $\alpha$ value increased, models better preserved the monotonically increasing trend, physically suggested by the data - i.e. the heuristic that MEQ damage that creates permeability is irreversible. A broad range of $\alpha$ values would result in models maintaining this monotonically increasing trend in permeability evolution.

## Data availability
The EGS Collab experiments data and seismic catalog data can be found at: https://gdr.openei.org/submissions/1311. Utah FORGE well 16 A (78)−32 stimulation injection data used in this study can be found at https://gdr.openei.org/submissions/1379. The seismic catalog recorded during Utah FORGE well 16 A (78)−32 stimulation can be access via https://gdr.openei.org/submissions/1429.

## Code availability
Codes supporting the findings of this manuscript are available from the corresponding author upon request.

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

## Acknowledgements

This work is a partial result of support from the US Department of Energy under grant DE-EE008763. This support is gratefully acknowledged. CJM acknowledges support from the European Research Council Advanced Grant 835012 (TECTONIC) the RETURN Extended Partnership and funding from the European Union Next-GenerationEU (National Recovery and Resilience Plan – NRRP, Mission 4, Component 2, Investment 1.3 – D.D. 1243 2/8/2022, PE0000005. PS acknowledges support from US Department of Energy (DE-SC0017585) and from the National Science Foundation under Grant No. (CMMI # 2121005). Any opinions, findings, and conclusions or recommendations expressed in this material are those of the author(s) and do not necessarily reflect the views of the National Science Foundation. DE gratefully acknowledges support from the US Department of Energy under grant EE0007080 through Utah FORGE, EGS-Collab funding through LBNL and the G. Albert Shoemaker endowment. The conclusions reported here are those of the authors.

## Author contributions

P.Y.: methodology, formal analysis, software, writing—original draft. A.M.: methodology, software, writing—review & editing. T.V.: software, investigation. A.B.: software, investigation. J.Y.: formal analysis. C.M.: methodology, writing—review and editing, supervision, project administration, and funding acquisition. P.S.: conceptualization, methodology, writing—review & editing, supervision, project administration. D.E.: conceptualization, methodology, formal analysis, writing—original draft, writing—review and editing, supervision, project administration, and funding acquisition. All authors contributed to the manuscript.

## Competing interests

The authors declare no competing interests.

## Additional information

**Supplementary information** The online version contains Supplementary Material available at https://doi.org/10.1038/s41467-024-46238-3.

