## [Peer Review File · Nature Communications]

Crustal Permeability Generated through Microearthquakes is Constrained by Seismic MomentREVIEWER COMMENTS

Reviewer #1 (Remarks to the Author):

This manuscript, which investigates the relationship between microearthquake magnitudes and permeability changes during hydraulic fracturing using a Bi-LSTM RNN model, presents a novel and significant approach in the field. The use of such advanced modeling to understand the dynamics of hydraulic fracturing is commendable. However, to enhance the clarity and robustness of the current work, several aspects require further elaboration and justification. While this study is promising, addressing the following points will enhance the manuscript's overall quality and comprehension. Here are the key areas for improvement:

1. Clarification of "High-Fidelity Data": On Page 3, you mention "high-fidelity data", but its definition and distinction from typical data remain unclear. A more comprehensive explanation would greatly benefit the reader's understanding.
2. Scientific Rationale for Data Selection: The choice of Ep3, Ep4, and Ep5 data in your analyses (Page 4) seems crucial. Please elaborate on the scientific reasoning behind this selection.
3. Representation of Spatiotemporal Distribution: The claim on Page 5 regarding the spherical distribution of the filtered MEQs appears somewhat arbitrary, especially when considering Fig. 4. A revision or a more convincing justification of this representation is suggested.
4. Explanation and Consistency of Equations: The derivation of Eqs. (2) and (3) on Page 6 needs to be more transparent. Including a conceptual diagram and a thorough explanation would be beneficial. Additionally, there seems to be a discrepancy in the dimensions of these equations, which requires clarification.
5. Justification of Delta_t_w Setting: The rationale behind setting Delta_t_w to 2 minutes (Page 7) is not provided. Including scientific reasoning for this choice would strengthen this section.
6. Choice of Bi-LSTM: An in-depth explanation for selecting Bi-LSTM for your analysis (Page 10) is necessary. Understanding the specific reasons for this choice would add depth to your methodology.
7. Determination of Alpha Values in Datasets: On Page 12, the basis for choosing alpha values of 100 and 150 in the EGS Collab and Utah FORGE datasets is not explained. Additional details on how these values were determined would be valuable.
8. Analysis of Prediction Accuracy: Finally, on Page 16, exploring the prediction accuracy related to permeability change and seismic rate or cumulative log seismic moment could be insightful. It's worth investigating if these relationships can be established without relying on deep learning.

Reviewer #2 (Remarks to the Author):

Review of NCOMMMS-23-56907, "Crustal Permeability Generated through Microearthquakes is Constrained by Seismic Moment" by Yu, P., Mail, A., Velaga, T., Bi, A., Yu, J., Marone, C., Shokouhi, P., and Elsworth, D.

General comments

This manuscript presents very interesting topic of constraining the magnitude of evolving crustal permeability via the cumulative seismic moment of microearthquakes (MEQ). I totally agree that the MEQ data could whisper some information of hydraulic structure and that the EGS Collab site or FORGE site could be the ideal sites to test this idea. Thus, the topic itself is very appropriate for *Nature Communications*. The authors claim that they partly success in predicting the time dependent changes in crustal permeability by combining MEQ data and machine learning model at both EGS Collab and FORGE sites. While I think the conclusions themselves are very impressive, I have the honest impression that there appears to be quite a few errors in the manuscript or figures, and as a result, many questions remain about the reliability of the conclusions. One important question is why the authors approximate MEQ cloud by a cylinder at EGS Collab site while they approximate it by a sphere at FORGE site. As far as the diffusion model is applied, the latter approximation should be consistently applied before proceeding with the analysis. In addition, I believe that the key point in this manuscript is the application of the deep learning (Bi-LSTM) model, but I think that the explanation of this point is inadequate. As it stands, the readers cannot follow the methodology exactly. With this in mind, at this point, I think it would be appropriate to reject the manuscript once and have it resubmitted after major revisions before publishing in *Nature Communications*. Specifically, I have the following comments:

1. Line 142-143: Why the authors approximate MEQ cloud by a cylinder at EGS Collab site while they approximate it by a sphere at FORGE site? This point should be clarified.
2. Line 158: It is not possible to determine from Figure 4 whether it is appropriate to approximate MEQ cloud with a sphere. Please revise Figure 4 to support this approximation.
3. Line 177: You should properly check the process of deriving the diffusion model (see Fluid-Induced Seismicity, 2015, Shapiro, S. A., Cambridge University Press).
4. Lines 180-181: How did you estimate or calculate hydraulic diffusivity, D ?
5. Lines 188 and 184: The dimensions should be properly checked. At least the dimensions are different in Eq (2) and Eq (3). Note the dimension of permeability is " m^2 ". Also, the terminologies "permeability", "conductivity", and "hydraulic diffusivity" should be properly distinguished.
6. Lines 216-217: I don't understand why this operation is appropriate. Wouldn't it be more

reasonable to consider a certain amount of these features?

7. Line 229-236: I think readers will only be confused if explanations using examples from other regions are suddenly developed here.
8. Fig 1 and Fig 3: A series of graphs are very difficult to read. For example, the legend is not visible. Also, I would like the vertical and horizontal axis scales to be standardized as much as possible. And, the units should be checked for permeability and seismic moment.
In addition, I do think that the relationship between the magnitude of each parameter is not consistent among Fig. 1, Fig. 3, and Fig. 15. For example, the cumulative seismic moment has a maximum at EP5 in Figure 1 but a maximum at EP3 in Figure 2, which is inconsistent. Because of these errors, the text cannot be fully trusted.
9. Line 283^{''}: I don't understand the essentials of the methodology. I would like to know what the inputs and outputs are and what you are trying to achieve with deep learning.
10. Line 382^{''}: Section 3.3 should be explained in the appendix.
11. Line 444 and 447: A proper distinction should be made between “permeability” and “change in the permeability”. The latter is usually expressed as follows:
$$A_k = k - k_0 = (b_0 + A_b)^{2/12} - b_0^{2/12}$$
12. Line 382^{''}: Section 3.3 should be explained in the appendix.
13. Line 465, 467, 478, and 480: Please clarify the definition of each parameter that appears in Equations (14)-(18). For example, I cannot find the difference between A_{k_n} and A_{k_s} .
14. Line 474: The approximation of $V^{''} a^3$ is not satisfactory. To make this approximation would mean that you think the aperture width is close to a. Is that correct?

I hope this helps.

We are grateful for the thoughtful comments that the reviewers have taken time to provide. We address these in the document below according to the following scheme:

Original reviewer comments are italicised.

Our direct response is in blue regular font.

Line XXX-XXX / Supplementary: “Changes to the text are quoted in yellow.”:

Reviewer #1 (Remarks to the Author):

This manuscript, which investigates the relationship between microearthquake magnitudes and permeability changes during hydraulic fracturing using a Bi-LSTM RNN model, presents a novel and significant approach in the field. The use of such advanced modeling to understand the dynamics of hydraulic fracturing is commendable. However, to enhance the clarity and robustness of the current work, several aspects require further elaboration and justification. While this study is promising, addressing the following points will enhance the manuscript’s overall quality and comprehension. Here are the key areas for improvement:

Thanks. We appreciate the time and effort spent to provide valuable feedback on our manuscript. Below, we provide our response to all the reviewer comments, where the corresponding changes to the manuscript are highlighted.

1. Clarification of "High-Fidelity Data": On Page 3, you mention "high-fidelity data", but its definition and distinction from typical data remain unclear. A more comprehensive explanation would greatly benefit the reader’s understanding.

Thanks for your comment on this point - we have clarified the designation of ‘High-fidelity’ in the revised manuscript. To illuminate the potential relationship between MEQs and induced permeability changes in hydraulic stimulations, high-quality datasets containing concurrent measurements of precise MEQ locations from high-resolution seismic networks together with local measurements of fluid injection pressures and volumes are all mandatory. Such data are very rare but EGS Collab and Utah FORGE datasets are two highly constrained field experiments that offer high-quality data that closely mimic those obtained during industrial stimulation efforts: that is, “high-fidelity” data. We now clarify this in the text.

Line 83-91: Illuminating the potential mechanistic relationship between MEQ characteristics and induced permeability changes requires access to high-quality datasets necessarily containing concurrent measurements of both quantities. These data should include accurate

MEQ locations from a high-resolution seismic network together with local measurements of fluid injection pressures and volumes. Such high-quality data are rare but a number of field trials are now available where permeability has been purposely created through hydraulic stimulation in the subsurface with concurrent seismic measurements. These highly constrained field experiments offer the possibility to retrieve the form of the relation linking features of the MEQs to the observed change in permeability using Machine Learning (ML) methods – we utilize these rare datasets.

2. Scientific Rationale for Data Selection: The choice of Ep3, Ep4, and Ep5 data in your analyses (Page 4) seems crucial. Please elaborate on the scientific reasoning behind this selection.

This is a good point. Injection rates for EGS Collab Ep1 and Ep2 are very low with few MEQs and miniscule changes in permeability (Fu et al., 2021; Schoenball et al., 2019) – thus there is no signal to effectively utilize in constraining the MEQ-permeability relationship. However, the Ep3, Ep4, and Ep5 observations are at much higher injection rates with many MEQs and with a discernible signal of permeability change. We revised the manuscript accordingly, as shown below:

Line 125-131: Five episodes of hydraulic stimulation were performed at EGS Collab in May 2018. The first two episodes (Ep1, and Ep2) used very low injection rates with few MEQs (Fu et al. 2021; Schoenball et al. 2019) and no permeability change signal to effectively utilize in constraining the MEQ-permeability relationship. However, we use data from the three subsequent continuous hydraulic stimulation episodes (Ep3, Ep4, and Ep5) where step-rate injections (Fig.1-A1, Fig.1-B1, Fig.1-C1) reactivated and created fractures adjacent to the injection borehole with significant signals for MEQs and permeability change.

Fu, P., Schoenball, M., Ajo - Franklin, J. B., Chai, C., Maceira, M., Morris, J. P., ... & EGS Collab Team. (2021). Close observation of hydraulic fracturing at EGS Collab Experiment 1: Fracture trajectory, microseismic interpretations, and the role of natural fractures. *Journal of Geophysical Research: Solid Earth*, 126(7), e2020JB020840.

Schoenball, M., Ajo-Franklin, J., Fu, P., & Templeton, D. (2019). Microseismic monitoring of meso-scale stimulations for the DOE EGS Collab project at the Sanford Underground Research Facility (No. LLNL-CONF-767025). Lawrence Livermore National Lab.(LLNL), Livermore, CA (United States).

3. Representation of Spatiotemporal Distribution: The claim on Page 5 regarding the spherical distribution of the filtered MEQs appears somewhat arbitrary, especially when considering Fig. 4. A revision or a more convincing justification of this representation is suggested.

Thanks for your comments. The original version of Fig.4 attempted to show the topology of all the events, which makes it difficult to illustrate the general spherical distribution of the MEQs. Therefore, we have modified Fig.4. It is always difficult to accurately characterize the MEQ

distribution using exact geometry. Here we assumed a simple spherical MEQ distribution as broadly characteristic of the flow geometry based on the MEQ observations. This assumption as spherical (UtahFORGE) or radial (EGS Collab) flow does not affect the process of linking MEQs to permeability changes since permeabilities calculated from the two geometries are linked to injectivity I (as the ratio of flow rate Q to pressure drop ΔP , $I = Q/\Delta P$) through a constant representing the flow geometry. We incorporate these points in the revised manuscript.

Thus, the most important issue in this regard is in distinguishing between the radial (EGS Collab) and spherical (Utah FORGE) propagation of the seismicity fronts used to define pressure boundary conditions in the different characterisations of permeability. We note the different forms of the flow systems identified in Eqs.(2) and (3) and their justification based on the seismic clouds and the flow geometries. Regardless of the choice of radial or spherical flow geometries, predicted permeabilities are insensitive to the exact geometry because permeability is linked to the ratio of flow rate to pressure drop through a single and nearly-invariant constant of proportionality.

Line 190-194: For EGS Collab, the seismic front (Fig. 2) propagates radially from the injection wellbore that is long in relation to the distal radius of the seismicity front, ultimately representing the external far-field pressure boundary. For FORGE, the injection zone length is short in comparison to the distal pressure boundary (Fig. 4) and evocative of spherical propagation of the seismicity front and corresponding far-field pressure boundary.

Line 204-209: We note that it is always difficult to accurately characterize the MEQ distribution using exact geometry. Here we assumed a simple spherical MEQ distribution as broadly characteristic of the flow geometry based on the MEQ observations. This assumption as spherical (Utah FORGE) or radial (EGS Collab) flow does not affect the process of linking MEQs to permeability changes since permeabilities calculated from the two geometries are linked to injectivity through a constant representing the flow geometry.

Line 285-290:

Fig.4. Location, timing, and magnitude of MEQs recorded during Utah FORGE stimulation tests during Stages 1-3 (S1-S3), where event timing is shown by symbol color with magnitude scaled by symbol radius. For the events shown, moment magnitude ranges from -2.09 to 0.52. Note spherical migration of the seismicity.

4. *Explanation and Consistency of Equations: The derivation of Eqs. (2) and (3) on Page 6 needs to be more transparent. Including a conceptual diagram and a thorough explanation would be beneficial. Additionally, there seems to be a discrepancy in the dimensions of these equations, which requires clarification.*

Thank you for pointing this out. There was a typo in Eq.(2) for the radial permeability calculation. The correct permeability equation for the EGS Collab experiment is: $k = \frac{\mu I}{2\pi h} \ln\left(\frac{r_t}{r_w}\right)$, which is also dimensionally consistent. The permeability calculations were previously completed with this correct relation, despite the typo in the text. We now include conceptual diagrams with associated derivations for Eqs.(2-3) in the Supplementary Information.

Line 196:

$$k = \frac{\mu I}{2\pi h} \ln\left(\frac{r_t}{r_w}\right). \quad (2)$$

Supplementary:

Figure S6 text

Fig. S8(a) illustrates the conceptual steady radial flow regime for constant injection rate, Q at differential pressure, P as:

$$\frac{Q}{2\pi r h} = \frac{k}{\mu} \frac{dP}{dr} \quad (S-1)$$

where r is the distance from the borehole wall, h is the length of the borehole section, k is permeability and μ is dynamic viscosity of the fluid. After variable separation and integration, we obtain:

$$\frac{Q}{2\pi h} \int_{r_w}^{r_t} \frac{dr}{r} = \frac{k}{\mu} \int_{P_e}^{P_d} dP \quad (S-2)$$

with average permeability could be expressed as:

$$k = \frac{\mu Q}{2\pi h(P_d - P_e)} \ln\left(\frac{r_t}{r_w}\right) \quad (S-3)$$

where P_d is the downhole injection pressure. P_e is the far-field pressure at the external radial boundary – defined by the location of the most distant MEQs. The injectivity (I) is the ratio between injection rate and pressure differential ($\Delta P = P_d - P_e$) as $I = Q/\Delta P$, Eq.(S-3) and may be further expressed as:

$$k = \frac{\mu I}{2\pi h} \ln\left(\frac{r_t}{r_w}\right) \quad (S-3)$$

Similarly, for the spherical steady flow shown in Fig.S8(b), the appropriate expression is:

$$\frac{Q}{4\pi R^2} = \frac{k dp}{\mu dr} \quad (S-4)$$

After variable separation and integration, Eq.(S-4) this becomes:

$$\frac{Q}{4\pi} \int_{r_w}^{r_t} \frac{dr}{r^2} = \frac{k}{\mu} \int_{P_e}^{P_d} dP \quad (S-5)$$

Thus the average permeability for spherical flow may be expressed as:

$$k = \frac{\mu Q}{4\pi(P_d - P_e)} \left(\frac{1}{r_w} - \frac{1}{r_t} \right) \quad (S-6)$$

And substituting injectivity (I) into Eq.(S-6) yields:

$$k = \frac{\mu I}{4\pi} \left(\frac{1}{r_w} - \frac{1}{r_t} \right) \quad (S-7)$$

Figure S8: Conceptual flow diagram of radial (a) and spherical (b) steady flow. Here r_w is the wellbore radius, r_t is the radius to the external flow boundary and h is the length of the borehole/cylindrical-zone.

5. Justification of Δt_w Setting: The rationale behind setting Δt_w to 2 minutes (Page 7) is not provided. Including scientific reasoning for this choice would strengthen this section.

Yes, good point. We use Δt_w to extract the features of seismicity rate in a moving time window that represents the response to injection. We plotted the seismicity rate changes under different Δt_w for both EGS Collab and Utah FORGE datasets, as shown in Supp. Fig. S4, Fig. S5. With increasing of Δt_w , the seismicity rate changes become more smoother, and some information of rapid seismic response is removed when using bigger Δt_w . Intuitively speaking,

an optimal result may exist since a short window (small Δt_w) misses the resulting change in permeability, consistent with the expectation that hydraulic response time is finite, and a too long window with large Δt_w smooths and smears the seismic response and effectively removes information from the data in the intensive hydraulic stimulation processes. We provide data for supplementary Bi-LSTM models under different Δt_w for both EGS Collab and Utah FORGE. As shown in Supp. Table S9, Table S10, the Bi-LSTM model with $\Delta t_w = 2 \text{ min}$ returns the best R^2 scores on both validation and test datasets although a range of values all return acceptable results. Interestingly, the R^2 score does not change significantly with increasing of Δt_w when $\Delta t_w > 2 \text{ min}$ especially for the Utah FORGE dataset as shown in Supp. Table S10. This insensitivity may reflect the fact that cumulative logarithmic seismic moment is the key feature in predicting permeability changes, and this feature does not change under different Δt_w . We add a short description of this in the manuscript and add detailed information and model results in the supplementary information to reflect this comment.

Line 239-244: This averaging approach, using a backward-looking moving time window of duration Δt_w , imposes a controllable degree of smoothing on the stochastic earthquake process (Supp. Fig. S4, Fig.S5). In this study, Δt_w was set as 2 minutes and the seismicity rate changes over time for both EGS Collab and Utah FORGE are depicted in Fig.1-(A4 to C4) and Fig.3-(A4 to C4), respectively. The rationale and sensitivity analysis of Δt_w is explored and discussed in Section 3.2.

Line 405-416: For the parameter Δt_w used for calculating seismicity rate feature, we studied the model performance under different Δt_w for both EGS Collab and Utah FORGE datasets. As shown in Supp. Table S9, Table S10, the Bi-LSTM model with $\Delta t_w = 2 \text{ min}$ returns the best R^2 scores on both validation and test datasets although a range of values all return acceptable results. An optimal result may exist since a short Δt_w misses the resulting change in permeability, consistent with the expectation that hydraulic response time is finite, and a long Δt_w smooths and smears the seismic response and effectively removes information from the data in the intensive hydraulic stimulation processes (Supp. Fig.S4, Fig.S5). Interesting, the R^2 does not change significantly with increasing of Δt_w (when $\Delta t_w > 2 \text{ min}$) especially for the Utah FORGE dataset as shown in Supp. Table S10. This insensitivity may reflect the fact that cumulative logarithmic seismic moment is the key feature in predicting permeability changes, and this feature does not change under different Δt_w .

Supplementary:

Figure S4: Seismicity rate changes over time under different Δt_w for EGS Collab dataset. With increases of Δt_w , the seismicity rate changes become much smoother for three episodes.

Figure S5: Seismicity rate changes over time under different Δt_w for Utah FORGE dataset. With increases of Δt_w , the seismicity rate changes become much smoother for three stages.

Table S9 Comparison of Bi-LSTM models using different values of Δt_w trained and tested on EGS Collab. All models were trained across 10 trials, with average R^2 scores and variances reported across data splits.

Δt_w (min)	Train (EP3) R^2	Validation (EP4) R^2	Test (EP5) R^2
2	0.99 ± 0.003	0.83 ± 0.07	0.93 ± 0.04

4	0.99 ± 0.00004	0.42 ± 0.003	0.74 ± 0.017
6	0.99 ± 0.0005	0.14 ± 0.00007	0.27 ± 0.006
8	0.96 ± 0.0015	0.19 ± 0.0002	0.65 ± 0.004
10	0.98 ± 0.0001	0.31 ± 0.0004	0.4 ± 0.06

Table S10 Comparison of Bi-LSTM models using different values of Δt_w trained and tested on Utah FORGE. All models were trained across 10 trials, with average R^2 scores and variances reported across data splits.

Δt_w (min)	Train (EP3) R^2	Validation (EP4) R^2	Test (EP5) R^2
2	0.99 ± 0.002	0.91 ± 0.005	0.85 ± 0.002
4	0.99 ± 0.00003	0.90 ± 0.0001	0.53 ± 0.00005
6	0.99 ± 0.00004	0.90 ± 0.0003	0.48 ± 0.00005
8	0.99 ± 0.000003	0.88 ± 0.0003	0.47 ± 0.0006
10	0.99 ± 0.00003	0.86 ± 0.0004	0.52 ± 0.0006

6. *Choice of Bi-LSTM: An in-depth explanation for selecting Bi-LSTM for your analysis (Page 10) is necessary. Understanding the specific reasons for this choice would add depth to your methodology.*

Yes. Thanks for your comment. We have added an explanation for selecting Bi-LSTM in the supplementary information, also included below:

Line 295-303: We note for our observations that stateful models such as LSTM are well suited for modelling sequential data and capturing the temporal dependence than stateless neural networks (Goodfellow et al., 2016). In ‘uni-directional’ LSTM models, the state at a given time captures the data history i.e., information in the preceding data samples. A bi-directional LSTM model is advantageous when the output (permeability change) depends on the entire predictor sequence (seismic moment) as it captures both backward and forward dependencies through time (Goodfellow et al., 2016). Additionally, bi-directional LSTM models provide improved stability and faster convergence, as detailed by analysis in the Supp. Table S11 and Text.

Goodfellow, I., Bengio, Y., & Courville, A. (2016). Sequence modeling: recurrent and recursive nets. Deep learning, 367-415.

Supplementary: Table S11; Table S11: Text

Table S11: Text

We are dealing with a stateful problem, which requires access to autoregressive connections. We selected two widely used RNNs, the Gated Recurrent Unit (GRU) and the Long-term Memory (LSTM) model. We observed that LSTM outperforms GRU in the majority of scenarios. Both uni-directional LSTM and bi-directional LSTM can efficiently model sequential data; however, a bi- model is advantageous when the output depends on the entire predictor sequence as it captures both backward and forward dependencies through time (Goodfellow et al., 2016, Stogin et al., 2020, Mali et al., 2023). In addition, the uni-directional model takes significantly more time to converge and is less stable compared to bi-directional. We compare an average number of epochs required by both models to converge on the EGS collab dataset (Table S11); as evident from the result, the bi-directional model converges much faster and is more stable with smaller standard deviation of R^2 than the uni-directional model.

Table S11: Comparison of LSTM and Bi-LSTM models trained on EGS collab. All models were trained across 10 trials, with average R^2 scores and variances reported on test set.

Model	Test (EP5) R^2	Average Epochs
Bi-LSTM	0.93 ± 0.04	700
LSTM	0.90 ± 0.09	2100

References:

- [1] Goodfellow, I., Bengio, Y., & Courville, A. (2016). Sequence modeling: recurrent and recursive nets. Deep learning, 367-415.
- [2] Stogin, J., Mali, A., & Giles, C. L. (2020). A provably stable neural network Turing Machine. arXiv preprint arXiv:2006.03651.
- [3] Mali, A., Ororbia, A., Kifer, D., & Giles, L. (2023). On the Computational Complexity and Formal Hierarchy of Second Order Recurrent Neural Networks. arXiv preprint arXiv:2309.14691.

7. Determination of Alpha Values in Datasets: On Page 12, the basis for choosing alpha values of 100 and 150 in the EGS Collab and Utah FORGE datasets is not explained. Additional details on how these values were determined would be valuable.

Yes, good point. The α parameter acts as a weighing factor for the added physics-based constraint to the loss function used to preserve the monotonically increasing trend in permeability during stimulation. This monotonic increase is physically justified since, once the reservoir is damaged by successive MEQs, the resulting change in permeability is additive and irrecoverable. A larger α gives more weight to the physics-based constraint and vice versa.

We have treated α as a tuneable parameter and searched for a value that provided well-performing models while sufficiently preserving the monotonically increasing trend. Figs. S6 and S7 compare the model performance on the test sets of EGS Collab and Utah FORGE across various α values. With an increase in α , the models better preserve the physically expected trend. Using such analyses, we chose α values of 1000 and 150 for the EGS Collab and Utah FORGE datasets, respectively. Note that there may be many α values that still preserve the monotonically increasing trend of prediction results.

Line 363-372: We used α values of 1000 and 150 for the EGS Collab and Utah FORGE datasets, respectively. We compared model performance between models using a proposed physically informed loss function and using a standard MES loss function ($\alpha = 0$) with results shown in Supp. Table S7 and Table S8 – these indicate that adding the physics-based constraint to the loss function aids convergence and better represents the relationship among different variables compared to standard regression loss. We also compared model performance across various α values (Suppl. Fig.S6, Fig.S7). As the α value increased, models better preserved the monotonically increasing trend, physically suggested by the data - i.e. the heuristic that MEQ damage that creates permeability is irreversible. A broad range of α values would result in models maintaining this monotonically increasing trend in permeability evolution.

Supplementary:

Figure S6: Comparison between raw permeability data (ground truth) and predictions on test set (Ep5) for EGS-Collab for different α values. With increasing α , the prediction curve on the test set monotonically increases. $\alpha = 1000$ is used in this study shown in Fig.6A of manuscript.

Figure S7: Comparison between raw permeability data (ground truth) and predictions on test set (S3) for Utah FORGE for different α values. With increasing α , the prediction curve on the test set monotonically increases. $\alpha = 150$ is used in this study shown in Fig.6B of manuscript.

8. *Analysis of Prediction Accuracy: Finally, on Page 16, exploring the prediction accuracy related to permeability change and seismic rate or cumulative log seismic moment could be insightful. It's worth investigating if these relationships can be established without relying on deep learning.*

That is a good question. The prediction accuracy of a model heavily relies on the distribution of the data across training, validation, and test sets as well as input features provided or learned by machine and deep learning models. Before deciding on deep learning models, we examined the performance of simpler (less complex) machine learning models. The simplest models, such as linear regression and even a single perceptron, could efficiently learn training distributions but failed when tested on validation and test distributions with an average R^2 score of 0.06, indicating extreme overfitting and failure to generalize. After extensive experiments, we observed that ensemble-based approaches such as XGBoost, and Voting Regression models can perform reasonably well; however, on-average, they are outperformed by deep learning models. For a detailed comparison of the performance of these models, please refer to Supplementary Tables S1 and S2.

Here is why we have relied on deep learning models in this study:

1. The distribution of train, validation, and test data is different, which requires models to exhibit “out-of-distribution” (OOD) generalization capability, and deep learning models generally perform better in such scenarios (Geirhos et al., 2020, Berend et al., 2020). As noted in the paper, we have introduced a novel normalization strategy to aid model prediction in our

OOD scenario (line 330-346). Without this strategy, even deep learning models would struggle to generalize.

2. Transfer learning: during transfer learning one needs to transfer learned parameters from source domain to target domain by assuming universality of feature space. We observe that transfer learning models converge faster (smaller epoch numbers) compared to standalone model and achieve comparable R^2 scores. Conversely, it is not possible to transfer representation for non-parametric Machine Learning model such as XGboost and Voting regression in their current formulation (Shokouhi et al., 2021).

Additionally, to elucidate the potential physical relationship between permeability changes and seismic moments, we present a theoretical model in the Discussion section (in the original and current version) of the manuscript. This model and related discussion defines the theoretical scaling linking seismic moment to permeability – to explain why the data-driven models perform well in this scenario.

Geirhos, R., Jacobsen, J. H., Michaelis, C., Zemel, R., Brendel, W., Bethge, M., & Wichmann, F. A. (2020). Shortcut learning in deep neural networks. *Nature Machine Intelligence*, 2(11), 665-673.

Berend, D., Xie, X., Ma, L., Zhou, L., Liu, Y., Xu, C., & Zhao, J. (2020, December). Cats are not fish: Deep learning testing calls for out-of-distribution awareness. In *Proceedings of the 35th IEEE/ACM international conference on automated software engineering* (pp. 1041-1052).

Shokouhi, P., Girkar, V., Rivière, J., Shreedharan, S., Marone, C., Giles, C. L., & Kifer, D. (2021). Deep learning can predict laboratory quakes from active source seismic data. *Geophysical Research Letters*, 48(12), e2021GL093187.

Line 446-454: Nevertheless, prediction accuracy of the model may be constrained by the quality of the dataset. This is particularly true for datasets that necessitate concurrent measurements of accurate MEQ locations from high-resolution seismic networks, along with local measurements of fluid injection pressures and volumes. For a relatively small dataset or when the relationship is simple, and all dataset splits come from the same distribution, one could observe machine learning models performing comparably to deep learning models; however, in the scenarios involving out-of-distribution and transfer learning, neural-based models often have an advantage over classical ML models (Geirhos et al., 2020, Berend et al., 2020, Shokouhi et al., 2021).

Reviewer #2 Evaluations:

Review of NCOMMMS-23-56907, "Crustal Permeability Generated through Microearthquakes is Constrained by Seismic Moment" by Yu, P., Mail, A., Velaga, T., Bi, A., Yu, J., Marone, C., Shokouhi, P., and Elsworth, D.

General comments:

This manuscript presents very interesting topic of constraining the magnitude of evolving crustal permeability via the cumulative seismic moment of microearthquakes (MEQ). I totally agree that the MEQ data could whisper some information of hydraulic structure and that the EGS Collab site or FORGE site could be the ideal sites to test this idea. Thus, the topic itself is very appropriate for Nature Communications. The authors claim that they partly success in predicting the time dependent changes in crustal permeability by combining MEQ data and machine learning model at both EGS Collab and FORGE sites. While I think the conclusions themselves are very impressive, I have the honest impression that there appears to be quite a few errors in the manuscript or figures, and as a result, many questions remain about the reliability of the conclusions. One important question is why the authors approximate MEQ cloud by a cylinder at EGS Collab site while they approximate it by a sphere at FORGE site. As far as the diffusion model is applied, the latter approximation should be consistently applied before proceeding with the analysis. In addition, I believe that the key point in this manuscript is the application of the deep learning (Bi-LSTM) model, but I think that the explanation of this point is inadequate. As it stands, the readers cannot follow the methodology exactly. With this in mind, at this point, I think it would be appropriate to reject the manuscript once and have it resubmitted after major revisions before publishing in Nature Communications. Specifically, I have the following comments:

We thank the Reviewer for a thoughtful review and evaluation for our manuscript. We have thoroughly revised the manuscript, addressed review comments and corrected errors. We address the three main concerns as: *G1*, *G2*, and *G3* and then the numbered comments.

G1: 'I have the honest impression that there appears to be quite a few errors in the manuscript or figures'

This point is similar to Question 8 (below), so we address it primarily below. And yes, sorry for the confusion, the short answer is that original Figs. 1 and 3 were not clear and too small to see. We had used scientific notation in Fig. 1 A4, B4, C4 for the cumulative seismic moment changes over time. We are sorry for this. We now plot these figures with improved clarity. The revised, clearer versions of Figs. 1 and 3 are also included below for reference.

Line 263-270:

Fig.1. Seismicity and injection observations for EGS Collab from Episode 3 (Ep3) to Episode 5 (Ep5). Top row (A1-C1) shows the evolution of injection pressures (P) and injection rates (Q) during hydraulic stimulation. Second row (A2-C2) shows the time history of injectivity (I) and MEQ moment magnitudes (M_w). Third row (A3-C3) shows the pressure-diffusive radius (r) fitted to the location of seismicity relative to the injection location (Shapiro et al., 1997; 2002). Fourth row (A4-C4) shows changes in permeability (k_c) and changes in two MEQ features (*viz.* seismicity rate (λ), and the cumulative log of seismic moment (\mathcal{M})).

Line 277-284:

Fig. 3. Seismicity and injection observations for the three stages of hydraulic stimulation at Utah FORGE. First row (A1-C1) shows the evolution of injection pressure (P) and injection rate (Q) during hydraulic stimulation. Second row (A2-C2) shows the time history of injectivity (I) and MEQ moment magnitudes (M_w). Third row (A3-C3) shows the pressure-diffusive radius (r) fitted to the location of seismicity relative to the injection location (Shapiro et al., 1997; 2002). Fourth row (A4-C4) shows permeability changes (k_c) and changes in two MEQ features (*viz.* seismicity rate (λ), and cumulative logarithm of seismic moment (\mathcal{M})).

G2: 'One important question is why the authors approximate MEQ cloud by a cylinder at EGS Collab site while they approximate it by a sphere at FORGE site. As far as the diffusion model is applied, the latter approximation should be consistently applied before proceeding with the analysis.'

Yes, please also see Reviewer #1 Q3 and Q4 responses.

The MEQ clouds for EGS Collab (Fig. 2) and Utah FORGE (Fig. 4) are of different form. The seismic front propagates radially outwards from the borehole for EGS Collab (Fig. 2) and spherically outwards from a shorter borehole section for Utah FORGE (Fig. 4). These two geometries thus define the geometry of the resulting flow systems that are created by the radially- or spherically-migrating seismic clouds and thus define the far-field pressure boundary as either a radial or spherical shell.

We have remade the figures to render the evolving seismic cloud geometry apparent, and explained, based on the morphology of this cloud, our choice of radial and spherical flow

models. In addition, we note that both radial and spherical flow models link the permeability to the ratio of flow rate (Q) and differential pressure (ΔP) through a constant of proportionality that is similar for both geometries.

As noted by your Q2 (below), the original version of Fig.4 tried to cover most of the events, which makes it difficult to illustrate the general spherical distribution of MEQs. Therefore, we modified Fig.4 as shown below with a refined smaller spherical distribution. It is always hard to accurately characterize the MEQ distribution as an exact geometry, and there could be other more optimal geometric approximations, however, that won't affect the whole process of linking MEQs to permeability changes since the flow geometry is just a scale factor ($\frac{\mu}{2\pi h} \ln\left(\frac{1}{r_w}\right)$ in Eq.2 and $\frac{\mu}{4\pi} \left(\frac{1}{r_w}\right)$ in Eq.3, respectively) that relates injectivity to permeability.

As for the diffusion model, we use that here merely to estimate the location of the seismic triggering front r_t in Eqs.2-3, which represents the separation between the MEQ migration front (or the pressure diffusion front) and the injection point or line source. It does not require a consistent MEQ cloud geometry (radial versus spherical) for both EGS Collab and Utah FORGE datasets, where different injection activities and operations were conducted.

Line 190-194: For EGS Collab, the seismic front (Fig. 2) propagates radially from the injection wellbore that is long in relation to the distal radius of the seismicity front, ultimately representing the external far-field pressure boundary. For FORGE, the injection zone length is short in comparison to the distal pressure boundary (Fig. 4) and evocative of spherical propagation of the seismicity front and corresponding far-field pressure boundary.

Line 204-209: We note that, it is always difficult to accurately characterize the MEQ distribution using exact geometry. Here we assumed a simple spherical MEQ distribution as broadly characteristic of the flow geometry based on the MEQ observations. This assumption as spherical (Utah FORGE) or radial (EGS Collab) flow does not affect the process of linking MEQs to permeability changes since permeabilities calculated from the two geometries are linked to injectivity through a constant representing the flow geometry.

Line 285-290:

Fig. 4. Location, timing, and magnitude of MEQs recorded during Utah FORGE stimulation tests during Stages 1-3 (S1-S3), where event timing is shown by symbol color with magnitude scaled by symbol radius. For the events shown, moment magnitude ranges from -2.09 to 0.52. Note spherical migration of the seismicity.

G3: *'In addition, I believe that the key point in this manuscript is the application of the deep learning (Bi-LSTM) model, but I think that the explanation of this point is inadequate. As it stands, the readers cannot follow the methodology exactly.'*

Response: Thanks for your observation/suggestion. We now explain the reasons for selecting the method and the application of Bi-LSTM. This is the same point as that raised by Reviewer 1, point 6 (please also see above for our reply and changes):

Line 295-303: We note for our observations that stateful models such as LSTM are well suited for modelling sequential data and capturing the temporal dependence than stateless neural networks (Goodfellow et al., 2016). In 'uni-directional' LSTM models, the state at a given time captures the data history i.e., information in the preceding data samples. A bi-directional LSTM model is advantageous when the output (permeability change) depends on the entire predictor sequence (seismic moment) as it captures both backward and forward dependencies through time (Goodfellow et al., 2016). Additionally, bi-directional LSTM models provide improved stability and faster convergence, as detailed by analysis in the Supp. Table S11 and Text.

Goodfellow, I., Bengio, Y., & Courville, A. (2016). Sequence modeling: recurrent and recursive nets. *Deep learning*, 367-415.

Supplementary: Table S11; Table S11: Text

Table S11: Text

We are dealing with a stateful problem, which requires access to autoregressive connections. We selected two widely used RNNs, the Gated Recurrent Unit (GRU) and the Long-term Memory (LSTM) model. We observed that LSTM outperforms GRU in the majority of scenarios. Both uni-directional LSTM and bi-directional LSTM can efficiently model sequential data; however, a bi- model is advantageous when the output depends on the entire predictor sequence as it captures both backward and forward dependencies through time (Goodfellow et al., 2016, Stogin et al., 2020, Mali et al., 2023). In addition, the uni-directional model takes significantly more time to converge and is less stable compared to bi-directional. We compare an average number of epochs required by both models to converge on the EGS collab dataset (Table S11); as evident from the result, the bi-directional model converges much faster and is more stable with smaller standard deviation of R^2 than the uni-directional model.

Table S11: Comparison of LSTM and Bi-LSTM models trained on EGS collab. All models were trained across 10 trials, with average R^2 scores and variances reported on test set.

Model	Test (EP5) R^2	Average Epochs
Bi-LSTM	0.93± 0.04	700
LSTM	0.90± 0.09	2100

References:

- [1] Goodfellow, I., Bengio, Y., & Courville, A. (2016). Sequence modeling: recurrent and recursive nets. *Deep learning*, 367-415.
- [2] Stogin, J., Mali, A., & Giles, C. L. (2020). A provably stable neural network Turing Machine. *arXiv preprint arXiv:2006.03651*.
- [3] Mali, A., Ororbia, A., Kifer, D., & Giles, L. (2023). On the Computational Complexity and Formal Hierarchy of Second Order Recurrent Neural Networks. *arXiv preprint arXiv:2309.14691*.

Numbered comments.

1. Line 142-143: Why the authors approximate MEQ cloud by a cylinder at EGS Collab site while they approximate it by a sphere at FORGE site? This point should be clarified.

Agreed. We explained this above in #G2 and in the reply to Reviewer 1.

2. Line 158: It is not possible to determine from Figure 4 whether it is appropriate to approximate MEQ cloud with a sphere. Please revise Figure 4 to support this approximation.

Thanks for your comments. The original version of Fig.4 attempted to include most of the events, which makes it hard to show that the general spherical distribution of the MEQs. Therefore, we modified Fig.4 as shown below with a refined smaller spherical distribution. As you know, it is always hard to accurately characterize the MEQ distribution using an exact geometry. Here we assumed a simple spherical MEQ distribution as approximately characteristic of the flow geometry based on observations. However, this assumption will not affect the process of linking MEQs to permeability changes since the flow geometry is just as a scale factor applied to the measured injectivity for calculating permeability. We incorporate these points in the revised manuscript.

Line 285-290:

Fig. 4. Location, timing, and magnitude of MEQs recorded during Utah FORGE stimulation tests during Stages 1-3 (S1-S3), where event timing is shown by symbol color with magnitude scaled by symbol radius. For the events shown, moment magnitude ranges from -2.09 to 0.52. Note spherical migration of the seismicity.

3. Line 177: You should properly check the process of deriving the diffusion model (see *Fluid-Induced Seismicity, 2015, Shapiro, S. A., Cambridge University Press*).

Thanks for your suggestion. We checked the process of deriving the diffusion model, and several related papers about diffusion models. We used the diffusion model only to estimate the location of the migrating seismic trigger front r_t , which is then used to calculate the permeability based on Eqs. 2 and 3 with the external boundary r_i defined. The first step is to obtain a best-fit hydraulic diffusivity D , identified in Question 4, following. Notably, we use this best fit merely to define the furthest extent of the seismicity cloud and hence as a boundary condition for the evaluation of the far-field pressure condition in the calculation of permeability from Eqs. (2) or (3).

Line 177-178: We use a diffusion model to follow the migration of the triggering front of the MEQ cloud (Shapiro et al., 1999, 2002, 2015).

Shapiro, S. A. (2015). *Fluid-induced seismicity*. Cambridge University Press.

4. Lines 180-181: How did you estimate or calculate hydraulic diffusivity, D ?

The hydraulic diffusivity D is estimated from the best-fit curve to the MEQ triggering front in the $r - t$ plot based on the diffusion equation (Shapiro, 2015), shown in Fig.1(A3, B3, C3), and Fig.3(A3, B3, C3). Then the best-fit curve defines the average spatial radius r_t or surface that separate two spatial domains, where the events will occur at distances that are smaller than or equal to r_t at a given time t (also represents the size of the relaxation zone (i.e. a spatial domain of significant changes) of the pore pressure) (Shapiro, 2015). The events are characterized by a significantly lower occurrence probability for larger distances than r_t at a given time t .

5. Lines 188 and 184: The dimensions should be properly checked. At least the dimensions are different in Eq (2) and Eq (3). Note the dimension of permeability is " m^2 ". Also, the

terminologies “permeability”, “conductivity”, and “hydraulic diffusivity” should be properly distinguished.

Thanks for pointing out this. There was indeed a typo in Eq.(2) for the approximated radial steady flow mean permeability calculation, that is now corrected. The correct relation was used in the prior calculations – that are unchanged. The correct permeability equation for radial flow (EGS Collab) is: $k = \frac{\mu l}{2\pi h} \ln\left(\frac{r_t}{r_w}\right)$.

Line 195:

$$k = \frac{\mu l}{2\pi h} \ln\left(\frac{r_t}{r_w}\right). \quad (2)$$

6. Lines 216-217: I don't understand why this operation is appropriate. Wouldn't it be more reasonable to consider a certain amount of these features?

Apologies for any confusion. We concur that incorporating a broader range of features for inputs in Deep Learning or Machine Learning models is a more reasonable approach. However, it is important to note that features manually derived and linked to the model output should not be utilized. In our case, we have already employed the locations of MEQs to determine the flow radius, based on the diffusion model, which was then utilized to calculate changes in permeability (as per the Bi-LSTM model output). Utilizing these as features (model inputs) might risk creating an artificial correlation between the MEQs location features and permeability changes. We have rephrased these sentences for better clarity.

Line 229-231: As the MEQ locations were used to define the evolving flow radius and to then calculate permeability, we did not extract features related to MEQ locations to avoid the risk of artificially correlating MEQs to permeability changes.

7. Line 229-236: I think readers will only be confused if explanations using examples from other regions are suddenly developed here.

Sorry for the confusion. We have re-phrased this paragraph.

Line 245-252: The second feature extracted is the cumulative logarithmic seismic moment. This metric has been instrumental in estimating the size of the activated reservoir volume, leading to the strategic placement of a new production well (Baisch et al., 2009; Rothert & Baisch, 2010). Moreover, a variety of field studies on seismicity triggered by fluid injection have shown that the total release of seismic energy (or seismic moment) is directly correlated with hydraulic energy (Bentz et al., 2020; Kwiatek et al., 2022). The cumulative logarithmic seismic moment,

\mathcal{M} , is defined as the cumulative sum of seismic moment during the cumulative time interval of $[0, t_i + \Delta t_w]$, expressed as:

8. Fig 1 and Fig 3: A series of graphs are very difficult to read. For example, the legend is not visible. Also, I would like the vertical and horizontal axis scales to be standardized as much as possible. And, the units should be checked for permeability and seismic moment.

In addition, I do think that the relationship between the magnitude of each parameter is not consistent among Fig. 1, Fig. 3, and Fig. 1S. For example, the cumulative seismic moment has a maximum at EP5 in Figure 1 but a maximum at EP3 in Figure 2, which is inconsistent. Because of these errors, the text cannot be fully trusted.

Response: We apologize for any confusion caused. We have revised and re-plotted Figures 1 and 3 to ensure clarity in both the vertical and horizontal axes, as well as the units for each variable. Figures 1 and S1 are now consistent with each other. In Figure 1 (panels A4, B4, C4), we have employed scientific notation to represent the cumulative seismic moment curves. For instance, in Figure 1, panels A4, B4, and C4, the scientific notation is indicated as $1e3$, $1e2$, and $1e2$ respectively for the cumulative log seismic moment curves. This denotes that the maximum cumulative log seismic moment occurs at Ep3, not Ep5, aligning with the data presented in Figure S1. The revised, clearer versions of Figures 1 and 3 are also attached below for reference.

Line 263-270:

Fig.1. Seismicity and injection observations for EGS Collab from Episode 3 (Ep3) to Episode 5 (Ep5). Top row (A1-C1) shows the evolution of injection pressures (P) and injection rates (Q) during hydraulic stimulation. Second row (A2-C2) shows the time history of injectivity (I) and MEQ moment magnitudes (M_w). Third row (A3-C3) shows the pressure-diffusive radius (r) fitted to the location of seismicity relative to the injection location (Shapiro et al., 1997; 2002). Fourth row (A4-C4) shows changes in permeability (k_c) and changes in two MEQ features (*viz.* seismicity rate (λ), and the cumulative log of seismic moment (\mathcal{M})).

Line 277-284:

Fig. 3. Seismicity and injection observations for the three stages of hydraulic stimulation at Utah FORGE. First row (A1-C1) shows the evolution of injection pressure (P) and injection rate (Q) during hydraulic stimulation. Second row (A2-C2) shows the time history of injectivity (I) and MEQ moment magnitudes (M_w). Third row (A3-C3) shows the pressure-diffusive radius (r) fitted to the location of seismicity relative to the injection location (Shapiro et al., 1997; 2002). Fourth row (A4-C4) shows permeability changes (k_c) and changes in two MEQ features (*viz.* seismicity rate (λ), and cumulative logarithm of seismic moment (\mathcal{M})).

9. Line 283-: I don't understand the essentials of the methodology. I would like to know what the inputs and outputs are and what you are trying to achieve with deep learning.

Thanks for your question. The inputs are the two features we extracted: the seismicity rate, and cumulative logarithmic seismic moment. The outputs are the permeability changes. We tried to use the features of the MEQs to predict the permeability changes during hydraulic stimulation

experiments using the datasets from the EGS Collab and Utah FORGE stimulation experiments. As noted in the manuscript (Lines 299-301), “The goal is to forecast permeability evolution (model output) from features of the MEQs (input features), namely seismicity rate, λ_i , and cumulative logarithmic seismic moment, \mathcal{M}_i extracted from the EGS Collab and Utah FORGE datasets”.

Regarding the methodology, we also explain the reasons for selecting the Bi-LSTM model. Please see the detailed explanation given above in #G2 and in the reply to Reviewer 1.

10. Line 382~: Section 3.3 should be explained in the appendix.

Fair enough, we moved this section to the Supplementary Information. Thank you!

11. Line 444 and 447: A proper distinction should be made between “permeability” and “change in the permeability”. The latter is usually expressed as follows: $\Delta k = k - k_0 =$

$$\frac{(b_0+\Delta b)^2}{12} - \frac{b_0^2}{12}$$

Yes. Agreed. We have revised this as below, and note that with the assumption that $\Delta b \gg b_0$ then the remainder of the scaling arguments and discussion still holds:

Line 478-483:

Changes in the permeability of individual fractures may be related to the change in aperture Δb from an original aperture b_0 as:

$$\Delta k = \frac{(b_0+\Delta b)^2}{12} - \frac{b_0^2}{12} \quad (9)$$

and to overall bulk permeability or transmissibility *via* the cubic law as (Ouyang & Elsworth, 1993):

$$\Delta k = \frac{(b_0+\Delta b)^3}{12s} - \frac{b_0^3}{12s} \quad (10)$$

12. Line 382~: Section 3.3 should be explained in the appendix.

Same comment as with Question 10 above. We moved this section to the Supplementary Information. Thank you!

13. Line 465, 467, 478, and 480: Please clarify the definition of each parameter that appears in Equations (14)-(18). For example, I cannot find the difference between Δk_n and Δk_s

Sorry for the confusion. We have added definitions of all these terms and include a Nomenclature section in the Supplementary Information for all the parameters included in the manuscript. Thanks for your comments.

Line 502: Where Δk_s is the permeability change resulting from fracture shearing.

Line 506: Where Δk_n is the permeability change resulting from fracture opening in extension.

Supplementary: Table S11: Nomenclature

a	Edge dimension of the area of the transected fault	M_0^s	Seismic energy released for slip
A	Fault patch area where slip occurred as $A \sim a^2$	M_0^n	Seismic energy released during fracture opening or closing
b_0	Original fracture aperture	\mathcal{M}	cumulative logarithmic seismic moment
b_s	Shear fracture aperture increment	P	Wellhead pressure
Δb	Fracture aperture change	Q	Flow rate
D	hydraulic diffusivity	r	Separation between the migrating seismic front and injection point
h	borehole/cylindrical-zone length	r_t	Radius to the migrating external flow boundary (defined by MEQ seismic front)
i	Fracture dilation angle	r_w	Injection wellbore radius
I	Injectivity	s	Spacing between adjacent parallel fractures
G	Shear modulus	Δt	elapsed time since initiation of injection t_0
k	Average permeability	Δt_w	Span of moving time window for seismicity rate calculation
k_c	Normalized permeability change	Δu_s	Fracture slip offset
k_0	Initial permeability	Δu_n	Normal displacement in fracture opening or closing
Δk	Permeability change compared to initial permeability	V	Volume of media surrounding the fault destressed by slip
Δk_s	Permeability change resulting from fracture shearing	α	Non-negative scalar penalty coefficient
Δk_n	Permeability change resulting from fracture tensile opening	μ	Fluid viscosity
\mathcal{L}	Adjusted loss function	λ	Seismicity rate

M_0	Seismic moment	$\Delta\tau$	Shear stress drop
M_w	Moment magnitude		

14. Line 474: The approximation of $V \sim a^3$ is not satisfactory. To make this approximation would mean that you think the aperture width is close to a . Is that correct?

We clarify this in the text to emphasize that b is aperture and a is the edge dimension of the transected fault prismatic which is of area A ($A = a^2$) and approximate tributary volume V ($V = a^3$) from which the strain energy of the seismic moment is derived (Kanamori and Anderson, 1975). If the fault is circular, rather than prismatic, then a constant is introduced to these trigonometric relations for area, A , and tributary volume, V , (Kanamori and Anderson, 1975), but including this analysis obscures the simplicity of the scaling relations – thus we opt for clarity.

Kanamori, H., & Anderson, D. L. (1975a). Theoretical basis of some empirical relations in seismology. *Bulletin of the Seismological Society of America*, 65(5), 1073–1095.

Line 511-513: Assuming this volume scales with the area of the transected fault of edge dimension a (Kanamori & Anderson, 1975) then for a prismatic fault area $A \sim a^2$ and $V \sim a^3$ enabling the scaling $A \sim V^{2/3}$ to be established.

I hope this helps.

We truly appreciate your comments.

REVIEWERS' COMMENTS

Reviewer #1 (Remarks to the Author):

As a reviewer of your manuscript, I would like to extend my appreciation for the thorough and appropriate manner in which you have addressed each of the review comments. It is evident that considerable effort and attention to detail were invested in responding to the feedback provided, ensuring that all concerns were effectively and satisfactorily resolved.

Reviewer #3 (Remarks to the Author):

This study utilizes microearthquake event rates and cumulative moments to assess changes in reservoir permeability induced by hydraulic fracturing. The improvement in permeability is linked to the rupture or opening of existing fractures and the generation of new fractures, leading to microseismicity. Therefore, the evaluation and prediction of reservoir permeability using microseismic features, such as seismic rates and cumulative moments, are theoretically feasible. The validity of this concept is demonstrated through the analysis of "High-Fidelity Data" from two Enhanced Geothermal System (EGS) projects.

After carefully reviewing the authors' responses to the previous comments and the revised manuscript, it appears that all concerns have been thoroughly addressed. Although I did not participate in the initial review, I have a few additional suggestions that I hope the authors will consider. I recommend proceeding with publication after addressing these points:

1. Basic Statistical Features of MEQ:

Provide a discussion on basic statistical features of the microseismic data, such as adherence to the Gutenberg-Richter relationship, completeness magnitude limit (M_c), and the b -value. Clarify how seismic events below M_c are treated in scenarios with different b -values.

2. Potential Fault Re-activation:

It's worth noting that hydraulic fracturing may activate nearby faults, leading to larger induced earthquakes (e.g., Ellsworth et al., GRL, doi: 10.1785/0220190102; Lei et al., Sci. China Earth Sci. 63, 1633-1660, doi: 10.1007/s11430-020-9646-x). In such cases, the relationship between injection volume (or fracturing volume) and cumulative seismic moment may no longer hold. This point should be discussed in the article. In the case of EGS-Collab in this study, there seems to be no such scenario. However, for FORGE, the filtering method used by the authors cannot distinguish earthquakes along penetrating faults (extending beyond the fracturing volume). In Figure 3A3-3C3, there are many events in the early stage outside the triggering front, exhibiting features of seismic cascading triggering. It is necessary to examine their impact on the training and prediction of the Bi-LSTM RNN model.

3. Microseismic Events Outside Triggering Front

In Figure 3A3-3C3, numerous events are observed outside the triggering front during the early stages, indicating features of seismic cascading triggering. Needs to evaluate the impact of these events on the training and prediction of the Bi-LSTM RNN model.

4. Hypocentre Symbols in Figures:

Resize hypocentre symbols in Figure 1-A3~1-C3 and Figure 3-A3~3-C3 based on M_w or M_o to enhance readability. Additionally, consider adding curves representing the back-front for better visualization.

5. Pore Pressure Decay and Microseismic Events

Clarify in lines 227-228 that microseismic events between the triggering front and back-front are also direct results of hydraulic fracturing, considering the decay law of pore pressure after the cessation of injection.

We are grateful for the thoughtful comments that the reviewers have taken time to provide. We address these in the document below according to the following scheme:

Original reviewer comments are italicised.

Our direct response is in blue regular font.

Line XXX-XXX / Supplementary: “Changes to the text are quoted in yellow.”:

Reviewer #1 (Remarks to the Author):

As a reviewer of your manuscript, I would like to extend my appreciation for the thorough and appropriate manner in which you have addressed each of the review comments. It is evident that considerable effort and attention to detail were invested in responding to the feedback provided, ensuring that all concerns were effectively and satisfactorily resolved.

Thanks a lot. We appreciate the time and effort spent to provide valuable feedback on our manuscript.

Reviewer #3 (Remarks to the Author):

This study utilizes microearthquake event rates and cumulative moments to assess changes in reservoir permeability induced by hydraulic fracturing. The improvement in permeability is linked to the rupture or opening of existing fractures and the generation of new fractures, leading to microseismicity. Therefore, the evaluation and prediction of reservoir permeability using microseismic features, such as seismic rates and cumulative moments, are theoretically feasible. The validity of this concept is demonstrated through the analysis of "High-Fidelity Data" from two Enhanced Geothermal System (EGS) projects.

After carefully reviewing the authors' responses to the previous comments and the revised manuscript, it appears that all concerns have been thoroughly addressed. Although I did not participate in the initial review, I have a few additional suggestions that I hope the authors will consider. I recommend proceeding with publication after addressing these points:

We thank the Reviewer for a thoughtful review and evaluation for our manuscript. Below, we have thoroughly revised the manuscript and addressed all review comments.

1. Basic Statistical Features of MEQ:

Provide a discussion on basic statistical features of the microseismic data, such as adherence to the Gutenberg-Richter relationship, completeness magnitude limit (M_c), and the b -value. Clarify how seismic events below M_c are treated in scenarios with different b -values.

Thanks for your comments. The magnitude of completeness is a common statistical way to determine the quality of an earthquake catalog and the detection capability of the seismic monitoring network. At EGS Collab, dense seismic networks were deployed for continuous passive and active seismic monitoring in the borehole, making it possible to generate the resulting high-quality catalog. For detailed information regarding the monitoring system, processing, quality control and quality evaluation of EGS Collab dataset, we refer in the manuscript to Schoenball et al.(2020). As for the Utah FORGE 16A(78)32 well hydraulic stimulation, an array of deep borehole geophone sensors was deployed as the primary network in monitoring wells 56-32, 58-32, and 78B-32 under the current maximum limit of 200 °C – for high resolution seismic monitoring. In addition, fiber-optic cables were also installed behind the casing in monitoring wells 78A-32 and 78B-32. For detailed information on the seismic monitoring systems at Utah FORGE, we point the reader to Rutledge et al. (2021).

Regarding the statistical features of the microseismic data for these projects, we analysed the completeness of magnitude (M_c) and b -value for both EGS Collab and Utah FORGE MEQs catalogs, as shown in Fig.S9. The maximum curvature method (Wiemer & Wyss, 2002) was used to calculate M_c , which is the maximum value of the first derivative of the frequency magnitude curve. The b -value is calculated according to the maximum likelihood method (Aki, 1965). We also used the uncertainty analysis for the b -value by calculating the standard error associated with the maximum likelihood method (Shi & Bolt, 1982). We obtained $M_c = -4.32$ for EGS Collab and -1.52 for Utah FORGE earthquake catalogs. Both these magnitudes are quite small, demonstrating the capability of the seismic networks to capture even small MEQs. Also, based on the linear proportionality relationship between permeability change and seismic moment, developed and noted in our Discussion section (Eqs.17-18), the effect of these extremely small magnitude events on permeability changes will be small – and missing these events is unlikely to impact our analysis or conclusions. The b -values for the EGS Collab and Utah FORGE are $b = 0.90 \pm 0.03$ and $b = 1.06 \pm 0.02$, respectively. Based on the Gutenberg-Richter relation, a b -value near 1.0 implies a typical distribution of earthquakes, with smaller events being more common than larger ones. We have added a short discussion on completeness of magnitude for the EGS Collab and Utah FORGE catalogs in the Supplementary file – noted below.

Schoenball, M., Ajo - Franklin, J. B., Blankenship, D., Chai, C., Chakravarty, A., Dobson, P., ... & EGS Collab Team. (2020). Creation of a mixed - mode fracture network at mesoscale through hydraulic fracturing and shear stimulation. *Journal of Geophysical Research: Solid Earth*, 125(12), e2020JB019807.

Rutledge, J., Pankow, K., Dyer, B., Wannamaker, P., Meier, P., Bethmann, F., & Moore, J. (2021). Seismic Monitoring at the Utah FORGE EGS Site. *GRC Transactions*, 45, 12.

Wiemer, Stefan, & Wyss, M. (2002). Mapping spatial variability of the frequency-magnitude distribution of earthquakes. In *Advances in geophysics* (Vol. 45, pp. 259–V). Elsevier.

Aki, K. (1965). Maximum likelihood estimate of b in the formula $\log N = a - bM$ and its confidence limits. *Bull. Earthq. Res. Inst., Tokyo Univ.*, 43, 237–239.

Shi, Yaolin, & Bolt, B. A. (1982). The standard error of the magnitude-frequency b value. *Bulletin of the Seismological Society of America*, 72(5), 1677–1687. <https://doi.org/10.1785/BSSA0720051677>

Supplementary: **Figure S9 text**

High resolution seismic monitoring networks were deployed for the EGS Collab and Utah FORGE stimulations. For detailed information regarding the monitoring system, processing, quality control and quality evaluation of the seismic catalogs and statistical parameters, please refer to Schoenball et al. (2020) and Rutledge et al. (2021). We analysed the magnitude of completeness (M_c) and b -values for the EGS Collab and Utah FORGE MEQ catalogs, as shown in Fig.S9. The maximum curvature method (Wiemer & Wyss, 2002) was used to calculate M_c , which is the maximum value of the first derivative of the frequency magnitude curve. The b -value is calculated by the maximum likelihood method (Aki, 1965). We also completed uncertainty analysis for the b -value by calculating the standard error associated with the maximum likelihood method (Shi & Bolt, 1982). We obtained $M_c = -4.32$ and -1.52 for EGS Collab and Utah FORGE earthquake catalogs, respectively. Both values are quite small, consistent with the capability of the seismic networks to capture even small MEQs. Also, based on the linear proportionality relationship between permeability change and seismic moment (see Discussion and Eqs.17-18), the effect of these extremely small magnitude events on permeability changes will be small – and missing these events is unlikely to impact our analysis or conclusions. The b -value for the EGS Collab and Utah FORGE are $b = 0.90 \pm 0.03$, $b = 1.06 \pm 0.02$ respectively. Based on the Gutenberg-Richter relation, a b -value near 1.0 implies a typical distribution of earthquakes, with smaller events being more common than larger ones.

Figure S9: Magnitude frequency distribution and Gutenberg-Richter fit with uncertainty analysis for EGS Collab (a) and Utah FORGE (b) datasets, respectively.

2. Potential Fault Re-activation:

It's worth noting that hydraulic fracturing may activate nearby faults, leading to larger induced earthquakes (e.g., Ellsworth et al., GRL, doi: 10.1785/0220190102; Lei et al., Sci. China Earth Sci. 63, 1633-1660, doi: 10.1007/s11430-020-9646-x). In such cases, the relationship between injection volume (or fracturing volume) and cumulative seismic moment may no longer hold. This point should be discussed in the article. In the case of EGS-Collab in this study, there seems to be no such scenario. However, for FORGE, the filtering method used by the authors cannot distinguish earthquakes along penetrating faults (extending beyond the fracturing volume). In Figure 3A3-3C3, there are many events in the early stage outside the triggering front, exhibiting features of seismic cascading triggering. It is necessary to examine their impact on the training and prediction of the Bi-LSTM RNN model.

Yes, just as you note, for larger earthquakes associated with fault reactivation, like the Pohang EGS stimulation Mw-5.5 event and several events larger than 5.0 in the Sichuan Basin, China during shale gas fracturing, it is possible that the relationship between injection volume and cumulative seismic moment may not hold. In our study, for the EGS Collab, the largest moment magnitude of events induced at EGS Collab is -1.83 (Fig.2) while the largest moment magnitude of events for Utah FORGE 16A (78)-32 well hydraulic stimulation is 0.52 (Fig.4). These MEQs are small and result from the creation of porosity with their locations hinting at the form and topology of the resultant architecture of connected permeable pathways. And we focus on linking the impacts of these many small MEQs to permeability enhancement during hydraulic stimulations, where the case of larger events induced due to fault reactivation may not exist. We added a short discussion about this in the manuscript.

The second part of the question is similar to comment #3, so we respond to it there.

Line 568-577: It is important to note that the relationship between injection volume and cumulative seismic moment may not always hold in the case of larger earthquakes induced by fault reactivation, such as the Mw-5.5 event during the Pohang EGS stimulation (Ellsworth et al., 2019; Li et al., 2021) and several events larger than 5.0 in the Sichuan Basin during shale gas fracturing (Lei et al., 2020). In our study, we focus on linking small MEQs to permeability enhancement during hydraulic stimulations. The largest moment magnitudes of events at the EGS Collab and Utah FORGE are -1.83 (Fig.2) and 0.52 (Fig.4), respectively. These small MEQs result from the creation of porosity, suggesting the form and topology of the resultant architecture of connected permeable pathways. Larger MEQs would also impact permeability in a similar way but large-scale reactivation of a discrete fault is not apparent in these field studies.

Ellsworth, W. L., Giardini, D., Townend, J., Ge, S., & Shimamoto, T. (2019). Triggering of the Pohang, Korea, earthquake (Mw 5.5) by enhanced geothermal system stimulation. *Seismological Research Letters*, 90(5), 1844-1858.

Li, Z., Elsworth, D., Wang, C., EGS-Collab (2021) Constraining maximum event magnitude during injection-triggered seismicity. *Nat. Commun.* 12:1528. <https://doi.org/10.1038/s41467-020-20700-4>

Lei, X., Su, J., & Wang, Z. (2020). Growing seismicity in the Sichuan Basin and its association with industrial activities. *Science China Earth Sciences*, 63, 1633-1660.

3. Microseismic Events Outside Triggering Front

In Figure 3A3-3C3, numerous events are observed outside the triggering front during the early stages, indicating features of seismic cascading triggering. Needs to evaluate the impact of these events on the training and prediction of the Bi-LSTM RNN model.

Thanks for your comment. Yes, some events are observed outside the triggering front during the early stages of stimulation – likely due to the impacts of stress projection beyond the fluid pressurization front (e.g. Bao and Eaton, 2016; Elsworth, et al., 2016). Nonetheless, we did not exclude these events; instead, we utilized all recorded events to delineate the features of the seismicity rate and cumulative seismic moment in our study. The migrating seismic triggering front represents the boundary between the relaxation zone of the pore pressure (pressure diffusion front) (Shapiro, 2015), which then could be used to calculate the permeability based on Eqs. 2 and 3 with the external boundary r_t defined. Once we recover the features of the MEQs (seismicity rate and cumulative seismic moment), and permeability changes, the Bi-LSTM model is used to evaluate their relationship. It might be interesting to investigate the impact of these few non-hydraulically connected events on the training and prediction of the Bi-LSTM model in a subsequent work. However, given the small number and magnitudes of these events, it is unlikely they significantly impact the performance of our Bi-LSTM model. This is due to the capability of the Machine Learning/Deep Learning models to link inputs to outputs by identifying both specific complex patterns and overarching trends within large datasets, even those containing noisy or irrelevant data (Mitchell, 1997; Goodfellow et al., 2016).

Mitchell, T. M. (1997). *Machine learning*.

Goodfellow, I., Bengio, Y., & Courville, A. (2016). *Deep learning*. MIT press.

Bao, X., Eaton, D.W. Fault activation by hydraulic fracturing in western Canada. *Science* 354, 1406-1409 (2016). DOI:10.1126/science.aag2583

Elsworth, D., Spiers, C.J., Niemeijer, A.R. (2016) Understanding induced seismicity. *Science* 354 (6318) pp. 1380–1381. <http://dx.doi.org/10.1126/science.aal2584>

4. Hypocentre Symbols in Figures:

Resize hypocentre symbols in Figure 1-A3~1-C3 and Figure 3-A3~3-C3 based on M_w or M_0 to enhance readability. Additionally, consider adding curves representing the back-front for better visualization.

Thanks for your suggestions, we plotted the events scaled by magnitude in Figures 1-a3 to 1-c3 and Figures 3-a3 to 3-c3. Our study focuses on the relationship between seismic moment and permeability changes during the period of hydraulic stimulations, so identifying the back-front of the induced seismicity, related to the period after the termination of fluid injection (Parotidis et al., 2004), seems less necessary – we respectfully choose to not include the back-front, to retain the clarity in the figure.

Parotidis, M., Shapiro, S. A., & Rothert, E. (2004). Back front of seismicity induced after termination of borehole fluid injection. *Geophysical research letters*, 31(2).

Line 257-263:

Fig.1. Seismicity and injection observations for EGS Collab from Episode 3 (Ep3) to Episode 5 (Ep5). Top row (a1-c1) shows the evolution of injection pressures (P) and injection rates (Q) during hydraulic stimulation. Second row (a2-c2) shows the time history of injectivity (I) and MEQ moment magnitudes (M_w). Third row (a3-c3) shows the pressure-diffusive radius (r) fitted to the location of seismicity relative to the injection location (Shapiro et al., 1997; 2002). Fourth row (a4-c4) shows changes in permeability (k_c) and changes in two MEQ features (*viz.* seismicity rate (λ), and the cumulative log of seismic moment (\mathcal{M})).

Line 271-278:

Fig. 3. Seismicity and injection observations for the three stages of hydraulic stimulation at Utah FORGE. First row (a1-c1) shows the evolution of injection pressure (P) and injection rate (Q) during hydraulic stimulation. Second row (a2-c2) shows the time history of injectivity (I) and MEQ moment magnitudes (M_w). Third row (a3-c3) shows the pressure-diffusive radius (r) fitted to the location of seismicity relative to the injection location (Shapiro et al., 1997; 2002). Fourth row (a4-c4) shows permeability changes (k_c) and changes in two MEQ features (viz. seismicity rate (λ), and cumulative logarithm of seismic moment (\mathcal{M})).

5. Pore Pressure Decay and Microseismic Events

Clarify in lines 227-228 that microseismic events between the triggering front and back-front are also direct results of hydraulic fracturing, considering the decay law of pore pressure after the cessation of injection.

Good point, clarified!

Line 218-220: We excluded MEQs recorded post-stimulation in this study, when events could have been triggered either by the diffusive expansion of fluid pressure decay (Parotidis et al., 2004) or poroelastic stressing (Segall & Lu, 2015).

Parotidis, M., Shapiro, S. A., & Rothert, E. (2004). Back front of seismicity induced after termination of borehole fluid injection. *Geophysical research letters*, 31(2).

Segall, P., & Lu, S. (2015). Injection - induced seismicity: Poroelastic and earthquake nucleation effects. *Journal of Geophysical Research: Solid Earth*, 120(7), 5082-5103.